# OPEL: Optimal Transport Guided ProcedurE Learning

**Sayeed Shafayet Chowdhury, Soumyadeep Chandra, and Kaushik Roy**
Elmore Family School of Electrical and Computer Engineering
Purdue University, West Lafayette, IN 47907, USA
`{chowdh23, chand133, kaushik}@purdue.edu`

## Abstract

Procedure learning refers to the task of identifying the key-steps and determining their logical order, given several videos of the same task. For both third-person and first-person (egocentric) videos, state-of-the-art (SOTA) methods aim at finding correspondences across videos in time to accomplish procedure learning. However, to establish temporal relationships within the sequences, these methods often rely on frame-to-frame mapping, or assume monotonic alignment of video pairs, leading to sub-optimal results. To this end, we propose to treat the video frames as samples from an unknown distribution, enabling us to frame their distance calculation as an optimal transport (OT) problem. Notably, the OT-based formulation allows us to relax the previously mentioned assumptions. To further improve performance, we enhance the OT formulation by introducing two regularization terms. The first, inverse difference moment regularization, promotes transportation between instances that are homogeneous in the embedding space as well as being temporally closer. The second, regularization based on the KL-divergence with an exponentially decaying prior smooths the alignment while enforcing conformity to the optimality (alignment obtained from vanilla OT optimization) and temporal priors. The resultant optimal transport guided procedure learning framework ('OPEL') significantly outperforms the SOTA on benchmark datasets. Specifically, we achieve 22.4% (IoU) and 26.9% (F1) average improvement compared to the current SOTA on large scale egocentric benchmark, EgoProceL. Furthermore, for the third person benchmarks (ProCeL and CrossTask), the proposed approach obtains 46.2% (F1) average enhancement over SOTA.

## 1 Introduction

The development of autonomous agents capable of reliably replicating human actions to accomplish certain end goals presents significant challenges. Traditional approaches would necessitate hard-coding tedious explicit instructions for each sub-task of the process (thus difficult to scale and generalize). A more efficient solution would involve the agent learning directly from observing multiple demonstrations of the assembly, without the need of any label. This motivates us to explore unsupervised procedure learning from videos. In the context of this work, procedure learning (PL) is conceptualized as the process of determining the key-steps and their correct sequential order to accomplish an overall task, as demonstrated across multiple video demonstrations [1, 2, 3].

PL analyzes multiple videos of a task as illustrated in Figure 1, in contrast to action-based tasks [4], which focus on a single video. The single-video approach is inapplicable to the identification of repetitive key-steps across videos. Moreover, action-based tasks typically neglect the sequencing of events, crucial for discerning an overall expected procedure composed of the sub-tasks. For instance, they fail to capture variations in the sequence of key-steps between videos V1 and V2 (Fig. 1). Other research efforts in video understanding that employ instructional videos include procedure planning [5], verifying sequences of procedures [6], and summarizing instructional content [7]. Additionally, unlike video alignment tasks [8], PL specifically aims to localize these essential steps within videos.

38th Conference on Neural Information Processing Systems (NeurIPS 2024).

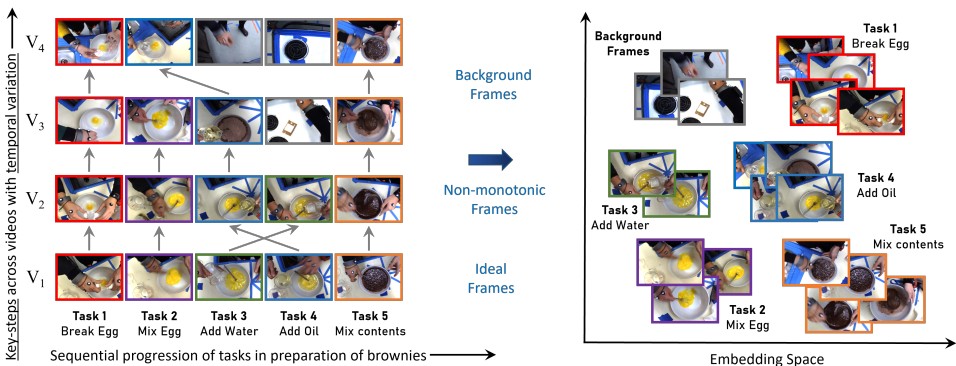

Figure 1: Key-steps required to prepare a brownie [17]. The sequences showcase temporal variations and corresponding key-step alignment challenges, namely (i) background frames (depicted as gray blocks), (ii) non-monotonic frames. OPEL aims to learn an embedding space where corresponding key-steps have similar embeddings while tackling the above challenges.

Much of the research on PL till now has been performed within the frameworks of supervised [9, 10, 11] and weakly supervised learning [12, 13, 14]. In a supervised setting, the reliance on per-frame annotations demands extensive manual labor. Conversely, weakly supervised methods involve using either ordered or unordered lists of key-steps. The generation of these lists requires either direct observation of the videos or specific heuristics, both of which pose significant scalability challenges [3]. Consequently, recent studies [1, 8] have shifted focus towards self-supervised learning, which do not require frame-wise labeling. Such a learning paradigm leverages the structured nature of accomplishing a complex task, which typically unfold in a predictable sequence of steps. For example, the act of preparing a "brownie" might involve breaking an egg, adding water, oil, mixing the contents, and then baking in the oven. The alignment of video frames is commonly performed in a monotonic manner [15], which presupposes a consistent order of actions across sequences. However, real-world sequences frequently deviate from this pattern, exhibiting temporal non-uniformities as depicted in Figure 1. These deviations can be categorized as follows: (i) background frames: frames irrelevant to the primary activity and should thus be excluded from alignment; (ii) redundant frames: these frames appear only in one sequence but not in others and do not contribute to the task; (iii) non-monotonic frames: these frames are characterized by a non-monotonic sequence of actions. Such frames challenge the assumption of monotonic progression and highlight the complexity of real-world activities. State-of-the-art (SOTA) methods adopt custom approaches to counter these irregularities such as removing background frames from processing [2], using extra information (e.g. gaze, depth) [16], or simply ignore them leading to suboptimal results [1].

To address the limitations of previous approaches, we relax the strict assumptions about the temporal sequence of actions and introduce a novel PL framework designed to learn temporal correspondences across videos. By treating instances of the sequences as samples from an unknown distribution, we formulate the task of computing the distance between them as an optimal transport (OT) [18] problem. The differentiable OT loss facilitates the alignment of non-monotonic sequences through frame-wise matching based on individual frame features. However, it typically overlooks the temporal smoothness and the inherent ordering relationships within the videos. To overcome this deficiency, we integrate two priors into the transportation matrix. First, the optimality prior favors the positions dictated by the OT, whereas the temporal prior discourages transport between temporally distant frames. Both these priors are modeled using a Laplace distribution with exponentially decaying probability from the corresponding centers. We introduce an additional virtual frame into the OT matrix to address background and redundant frames. Furthermore, to avoid the common issue of converging to trivial solutions in temporal video alignment [15], we employ a novel inter-video contrastive loss, which acts as a regularizer. Finally, the sub-tasks of each video are clustered in the embedding space using graphcut segmentation [19]. The overall framework, termed optimal transport guided procedure learning ('OPEL'), achieves SOTA results on both the ego and exocentric benchmarks. To summarize, our main contributions are-

- We propose a novel optimal transport based procedure learning framework that aligns frames with similar semantics together in an embedding space.

- To enhance the OT-based learning, we integrate optimality and temporal priors, both modeled using the Laplace distribution. These two priors also serve as regularizers. Furthermore, OPEL incorporates a novel inter-video contrastive loss for additional improvement.
- OPEL demonstrates substantial performance gains, achieving an average improvement of 22.4% in IoU and 26.9% in F1-score compared to the current SOTA on the EgoProceL benchmark.

## 2   Related Works

**Representation Learning for Videos.** Recent studies have explored various pretext tasks to facilitate representation learning through self-supervised or unsupervised approaches. Examples include temporal coherence and sequence ordering [20, 21, 22, 23, 24], predicting frames [25, 26, 27, 28, 29], and determining the directionality of time [30]. These methods typically derive signals from a constrained set of videos. In contrast, our objective is to discern and characterize key-steps of a certain task across multiple videos, expanding the scope and applicability of representation learning.

**Self-Supervised Representations for Procedure Learning.** Previous studies on PL have focused on developing methods for learning frame-level features [31, 3, 32, 33]. For instance, Kukleva et al. [32] enhance the representation space by utilizing relative timestamps of frames, while Vidal et al. [33] engage in predicting future frames along with their timestamps. Elhamifar et al. [3] apply attention mechanisms to individual frames to enhance feature learning. Similarly, Bansal et al. [1] leverage temporal correspondences across videos to generate signals and learn frame-level embeddings. The current SOTA model for egocentric PL [2] utilizes task-level graph representation to cluster semantically similar and temporally close frames. Despite these advancements, these methods often exhibit limitations in adequately modeling either temporal or spatial relationships within video sequences, especially in the presence of background and redundant frames. As a result, extra curated processing steps are required, resulting in additional layer of complexity and computation e.g. [2] depends on background frame removal to improve performance.

**Multi-modal Procedure Learning.** PL has also been used with multi-modal data such as (a) narrated text and videos [34, 35, 36, 37, 38, 39], (b) optical flow, depth and gaze information [16]. These studies typically rely on the assumption of a reliable alignment between video content and corresponding supporting modalities [34, 38, 39], an assumption that often proves inaccurate [31, 3] due to lack of synchrony among the modalities. Additionally, the dependence on imperfect Automatic Speech Recognition (ASR) systems necessitates subsequent manual corrections. Moreover, multiple modalities require additional memory and compute. In contrast, our framework exclusively leverages visual data, thereby circumventing the inaccuracies associated with multimodal alignment and enhancing scalability by eliminating the need for extra data modalities.

**Video Alignment.** can be efficiently addressed in synchronized settings using established methods like Canonical Correlation Analysis (CCA) [40] and soft-Dynamic Time Warping (DTW) [41]. A recent work [42] aligns videos by learning self-supervised representations from multiple viewpoints. However, the requirement for synchronized multi-view recordings limits its applicability. To tackle this challenge, [8] proposes a cycle consistency loss to establish frame correspondences, focusing primarily on local matches and not on the global temporal structure of the videos. Perhaps, works leveraging OT for visual analysis [43, 44, 45] are most related to our approach. But, such approaches do not address sequence alignment as we do. An exception is [46], which employs OT for videos, however their setup for evaluation is supervised fine-tuning for action segmentation, a fundamentally different task than unsupervised PL. As a result, [46] does not deal with temporal localization of the key-steps of a task nor their ordering, unlike us. Moreover, our modeling of priors using Laplace distribution and inter-video contrastive loss formulation are different from [46].

**Learning Key-step Ordering.** Most existing studies fail to account for variations in the ordering of key-steps required to complete a task, often assuming either a strict sequential order [31, 32, 33] or neglecting to model the sequence altogether [3, 47]. However, as illustrated in Figure 1, individuals frequently execute the same task in diverse manners, underscoring the need for a more flexible approach. To that end, OPEL is designed to identify and construct a unique key-step sequence for each video, thereby adapting to and inferring the specific ordering of the task.

## 3   OPEL Framework

**Optimal Transport Formulation.** OT provides a metric for assessing the dissimilarity between two probability distributions within a metric space [18]. By using the feature vectors from each entity and a distance matrix between them, it establishes correspondences that minimize total distance, also

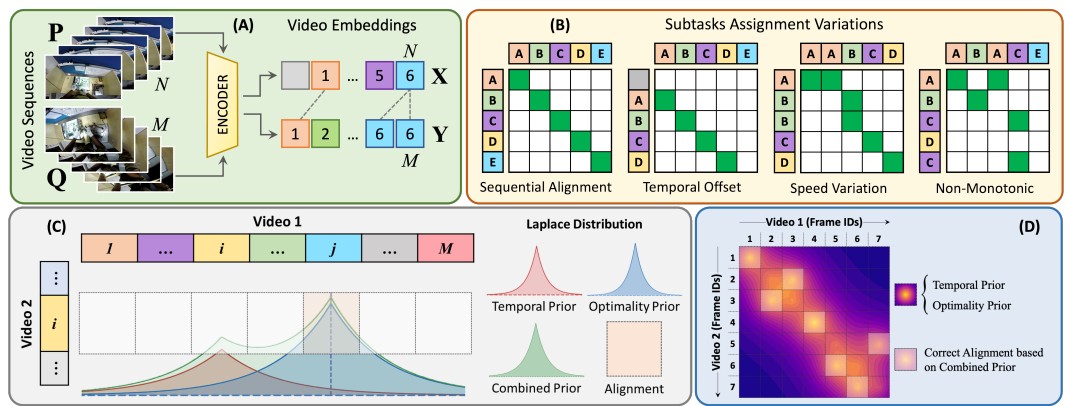

Figure 2: (A) The encoder generates frame-wise embeddings from videos, facilitating subsequent OT calculations. (B) Pair-wise scenarios captured through the assignment matrix- from strictly synchronized actions to temporal shifts and differing action speeds, to non-monotonicity. (C) 1-D depiction of alignment of a single frame ($i$-th) of Video 2 with its best match frame ($j$-th) of Video 1, based on the proposed priors. (D) 2-D representation of the optimal alignment of frame sequences.

ensuring optimality, separability, and completeness. Assume, the inputs are two sequences of video frames, $\boldsymbol{P} = [\boldsymbol{p}_1, \boldsymbol{p}_2, \ldots, \boldsymbol{p}_N]$ and $\boldsymbol{Q} = [\boldsymbol{q}_1, \boldsymbol{q}_2, \ldots, \boldsymbol{q}_M]$. We pass these through a deep encoder network (as illustrated in Fig. 2(A)) to obtain their respective embeddings, $\boldsymbol{X} = [\boldsymbol{x}_1, \boldsymbol{x}_2, \ldots, \boldsymbol{x}_N]$ and $\boldsymbol{Y} = [\boldsymbol{y}_1, \boldsymbol{y}_2, \ldots, \boldsymbol{y}_M]$. Let, $(\Omega, l)$ is a metric space, where $l : \Omega \times \Omega \to \mathbb{R}$ denotes the distance in $\Omega$, and $P(\Omega)$ represents all Borel probability measures on $\Omega$. Considering the elements of $\boldsymbol{X}$ and $\boldsymbol{Y}$ as independent samples, their probability measures can be written as, $f = \sum_{i=1}^{N} \alpha_i \delta_{x_i}$ and $g = \sum_{j=1}^{M} \beta_j \delta_{y_j}$, where $\delta_x$ denotes the Dirac mass at $x$, and $\boldsymbol{\alpha}$ and $\boldsymbol{\beta}$ are the weights for the distributions $f$ and $g$, respectively. Since there is no justification for assigning greater importance to one frame over another, initially we set $\alpha_i = \frac{1}{N}$ and $\beta_j = \frac{1}{M}$ for all $i, j$, leading to a feasible set of weight matrices defined as the transportation polytope [48], $U(\boldsymbol{\alpha}, \boldsymbol{\beta}) := \{\boldsymbol{T} \in \mathbb{R}_+^{N \times M} : \boldsymbol{T} \boldsymbol{1}_M = \boldsymbol{\alpha}, \boldsymbol{T}^\top \boldsymbol{1}_N = \boldsymbol{\beta}\}$. Here, $t_{ij}$ can be interpreted to be proportional to the probability that $\boldsymbol{x}_i$ will be aligned to $\boldsymbol{y}_j$. We start by computing the pairwise Euclidean distances between embedding vectors, $d(\boldsymbol{x}_i, \boldsymbol{y}_j) = \|\boldsymbol{x}_i - \boldsymbol{y}_j\|$ to form the $N \times M$ distance matrix, $\boldsymbol{D}$. The cost of transporting mass from $f$ to $g$ with a transport plan $\boldsymbol{T}$ is quantified by the Frobenius inner product $\langle \boldsymbol{T}, \boldsymbol{D} \rangle$. Thus, the Wasserstein distance raised to the power $p$ is: $W_p^p(f, g) = l_W(\boldsymbol{\alpha}, \boldsymbol{\beta}, \boldsymbol{D}) = \min_{\boldsymbol{T} \in U(\boldsymbol{\alpha}, \boldsymbol{\beta})} \langle \boldsymbol{T}, \boldsymbol{D} \rangle$. We only consider $p = 1$, and drop $p$ henceforth. To simplify the above optimization and make training feasible, Cuturi [48] introduced an entropy regularization, leading to the Sinkhorn distance,

$$l_\lambda^S(\boldsymbol{\alpha}, \boldsymbol{\beta}, \boldsymbol{D}) = \langle \boldsymbol{T}_\lambda, \boldsymbol{D} \rangle \quad \text{s.t. } \boldsymbol{T}_\lambda = \arg \min_{\boldsymbol{T} \in U(\boldsymbol{\alpha}, \boldsymbol{\beta})} \langle \boldsymbol{T}, \boldsymbol{D} \rangle - \frac{1}{\lambda} h(\boldsymbol{T}), \tag{1}$$

where $h(\boldsymbol{T}) = -\sum_{i=1}^{N} \sum_{j=1}^{M} t_{ij} \log t_{ij}$ denotes the entropy of $\boldsymbol{T}$, and $\lambda$ is the regularization parameter. The optimal solution for Eqn. (1) has the form [48], $\boldsymbol{T}_\lambda = \text{diag}(\boldsymbol{\kappa}_1) \exp(-\lambda \boldsymbol{D}) \text{diag}(\boldsymbol{\kappa}_2)$, where $\exp(-\lambda \boldsymbol{D})$ is the element-wise exponential of the matrix $-\lambda \boldsymbol{D}$, and $\boldsymbol{\kappa}_1 \in \mathbb{R}^N$ and $\boldsymbol{\kappa}_2 \in \mathbb{R}^M$ are the non-negative left and right scaling vectors to be obtained by the Sinkhorn fixed point iterations.

**Regularization with Priors.** The above formulation minimizes the cost of aligning two sequences, however it totally neglects the temporal ordering relationships inherent in videos, failing to leverage the temporal consistency. Typically, the alignment of multiple videos depicting the same activity should constrain the temporal position of one sequence to correspond closely with adjacent temporal positions of another sequence. Perfect alignment would render the transport matrix $\boldsymbol{T}$ diagonal, but this strict requirement is impractical for real-world applications. As illustrated in Fig. 2(B), variations such as earlier commencement of activities, differing action speeds, or non-monotonic sequences complicate alignment. To address these challenges and achieve optimal alignment while accounting for temporal variations, we introduce two priors into the OT framework. Essentially, there are 2 factors in play - (i) optimality which tries to find the best match between frames irrespective of their temporal distance (which may result in temporally incoherent alignment), and (ii) the temporal factor which promotes transport between nearby frames only without considering their feature matching. We hypothesize that the optimal solution requires striking a balance between both, and thus propose to enhance the OT formulation by incorporating two specific priors addressing the above factors [46].

The first prior, termed the 'Optimality Prior', is introduced to effectively manage non-monotonic sequences. This prior leverages the transport matrix $\boldsymbol{T}$ as delineated in Eqn. (1), which provides a preliminary indication of alignment between two video sequences. This matrix adapts dynamically to reflect the temporal variations observed across the sequences. Our approach uses this dynamic behavior to establish the optimality prior. We want the point representing the most likely alignment according to $\boldsymbol{T}$ to have the highest likelihood, while the assignment probability decays along any perpendicular direction from this center. Specifically, we model this as a Laplace distribution,

$$\boldsymbol{Q}_o(i,j) = \frac{1}{2b} e^{-\frac{|d_o(i,j)|}{b}}, \quad \text{where } d_o(i,j) = \frac{|i/N - i_o/N| + |j/M - j_o/M|}{2\sqrt{1/N^2 + 1/M^2}}, \tag{2}$$

represents the average distance from $(i,j)$ to the frame locations $(i, j_o)$ and $(i_o, j)$ that correspond to the optimal alignment as indicated by the transport matrix, and $b$ is a scale parameter. Motivated by [46], we incorporate a second prior, termed the 'Temporal Prior', which promotes alignment of one sequence with elements in proximal temporal positions of the other sequence, thereby preserving the overall temporal structure and maintaining consistency in action order. This prior results in an assignment matrix characterized by peak values along the diagonal, with values diminishing perpendicular to the diagonal. Again, this scenario is modeled using a two-dimensional Laplace distribution, where the distribution along any line perpendicular to the diagonal is exponentially decaying, centered along the diagonal itself:

$$\boldsymbol{Q}_t(i,j) = \frac{1}{2b} e^{-\frac{|d_t(i,j)|}{b}}, \quad \text{where } d_t(i,j) = \frac{|i/N - j/M|}{\sqrt{1/N^2 + 1/M^2}} \tag{3}$$

is the distance from $(i,j)$ to the diagonal. Inspired by [46], we merge these priors as,

$$\boldsymbol{Q}(i,j) = \phi \, \boldsymbol{Q}_t(i,j) + (1 - \phi) \, \boldsymbol{Q}_o(i,j), \tag{4}$$

where, $\phi$ serves as a dynamic weight, initially set to 1.0, and progressively reduced to 0.5 during training. This gradual adjustment of $\phi$ allows the model to adaptively improve its alignment based on the increasing fidelity of the OT predictions. Optimal alignment based on these priors is pictorially depicted in Fig.2(C, D). Note, the 1-dimensional alignment in Fig.2(C) is for demonstration only, our actual implementation is based on 2-dimensional distributional priors as shown in Fig.2(D).

**Background and Redundant Frames.** To effectively manage background and redundant frames, we integrate an additional 'virtual frame' within the transport matrix, following [46]. This serves as a placeholder for aligning any frame that do not match with the primary sequence, and allows OPEL to explicitly assign these non-contributing frames to the virtual frame, as shown in Fig. 4(B). The augmented transport matrix, now denoted as $\hat{\boldsymbol{T}} \in \mathbb{R}^{(N+1)\times(M+1)}$, includes an extra row and column to accommodate the virtual frame. Note, if the likelihood of a frame aligning with any salient frame falls below a predefined threshold, $\zeta$, we assign that frame to the virtual frame.

**Training Methodology.** While the above formulation sounds promising, devising a differentiable framework to leverage these during training is pivotal. To that effect, following [46], we define 2 terms to effectively regularize $\hat{\boldsymbol{T}}$. To capture the essence of $\boldsymbol{Q}_o$, $\hat{\boldsymbol{T}}$ needs to be structured to highlight prominent values at locations corresponding to the most probable alignments,

$$M_o(\hat{\boldsymbol{T}}) = \sum_{i=1}^{N+1} \sum_{j=1}^{M+1} \frac{t_{ij}}{\frac{1}{2} d_m + 1}, \quad \text{where } d_m = \left( \frac{i - i_o}{N+1} \right)^2 + \left( \frac{j - j_o}{M+1} \right)^2 \tag{5}$$

Similarly, to conform to $\boldsymbol{Q}_t$, $\hat{\boldsymbol{T}}$ is expected to exhibit prominent values along its diagonal, reflecting temporally closely aligned frames; while off-diagonal elements should ideally possess diminished magnitudes. This sort of structural arrangement can be quantitatively assessed using:

$$M_t(\hat{\boldsymbol{T}}) = \sum_{i=1}^{N+1} \sum_{j=1}^{M+1} \frac{t_{ij}}{\left( \frac{i}{N+1} - \frac{j}{M+1} \right)^2 + 1} \tag{6}$$

Similar to Eqn. (4), the above 2 equations are combined as a regularizer on the transport matrix, $M(\hat{\boldsymbol{T}}) = \phi \, M_t(\hat{\boldsymbol{T}}) + (1 - \phi) \, M_o(\hat{\boldsymbol{T}})$. Such a structure is known as the inverse difference moment (IDM) [49, 46]. To encourage optimal alignment, $M(\hat{\boldsymbol{T}})$ of the learned $\hat{\boldsymbol{T}}$ should be maximized. In order to facilitate this and to ensure the smooth assignment of such matches, we define a modified feasible set for $\hat{\boldsymbol{T}}$ by incorporating two additional constraints into the set $U(\boldsymbol{\alpha}, \boldsymbol{\beta})$,

$$U_{\xi_1, \xi_2}(\boldsymbol{\alpha}, \boldsymbol{\beta}) = \left\{ \hat{\boldsymbol{T}} \in \mathbb{R}_+^{N+1 \times M+1} \mid \hat{\boldsymbol{T}} \mathbf{1}_{M+1} = \boldsymbol{\alpha}, \hat{\boldsymbol{T}}^\top \mathbf{1}_{N+1} = \boldsymbol{\beta}, M(\hat{\boldsymbol{T}}) \geq \xi_1, \text{KL}(\hat{\boldsymbol{T}} \parallel \hat{\boldsymbol{Q}}) \leq \xi_2 \right\} \tag{7}$$

where $\text{KL}(\hat{\boldsymbol{T}} \parallel \hat{\boldsymbol{Q}}) = \sum_{i=1}^{N+1} \sum_{j=1}^{M+1} t_{ij} \log \frac{t_{ij}}{q_{ij}}$ is the Kullback- Leibler (KL) divergence between $\hat{\boldsymbol{T}}$ and $\hat{\boldsymbol{Q}}$, and $\hat{\boldsymbol{Q}}$ is same as Eqn. (4) but augmented with the virtual frame. So, the regularized Wasserstein distance between $\boldsymbol{X}$ and $\boldsymbol{Y}$ now becomes -

$$l_{\xi_1,\xi_2}^R(\boldsymbol{X},\boldsymbol{Y}) = \min_{\hat{\boldsymbol{T}} \in U_{\xi_1,\xi_2}(\boldsymbol{\alpha},\boldsymbol{\beta})} \langle \hat{\boldsymbol{T}}, \boldsymbol{D} \rangle. \tag{8}$$

The above optimization can be efficiently solved by considering its dual. As such, we incorporate two Lagrange multipliers, $\lambda_1 > 0$ and $\lambda_2 > 0$, to obtain the dual of Eqn. (8) as-

$$l_{\lambda_1,\lambda_2}^R(\boldsymbol{X},\boldsymbol{Y}) := \langle \hat{\boldsymbol{T}}_{\lambda_1,\lambda_2}, \boldsymbol{D} \rangle, \text{ s.t. } \hat{\boldsymbol{T}}_{\lambda_1,\lambda_2} = \arg \min_{\hat{\boldsymbol{T}} \in U(\boldsymbol{\alpha},\boldsymbol{\beta})} \langle \hat{\boldsymbol{T}}_{\lambda_1,\lambda_2}, \boldsymbol{D} \rangle - \lambda_1 M(\hat{\boldsymbol{T}}) + \lambda_2 \text{KL}(\hat{\boldsymbol{T}} \parallel \hat{\boldsymbol{Q}}). \tag{9}$$

The optimal $\hat{\boldsymbol{T}}_{\lambda_1,\lambda_2}$ that optimizes Eqn. (9) is $e^{diag(-\frac{1}{2} - \frac{\boldsymbol{\mu}}{\lambda_2})} \boldsymbol{K} e^{diag(-\frac{1}{2} - \frac{\boldsymbol{\nu}}{\lambda_2})}$, where $\boldsymbol{K} = [q_{ij} e^{\frac{1}{\lambda_2}(s_{ij}^{\lambda_1} - d_{ij})}]_{ij}$, $s_{ij}^{\lambda_1} = \lambda_1 \left( \frac{1}{(\frac{i}{N+1} - \frac{j}{M+1})^2 + 1} + \frac{1}{\frac{1}{2}d_m + 1} \right)$, $d_m$ given by Eqn.(5), and $\boldsymbol{\mu}$ and $\boldsymbol{\nu}$ are the dual variables for the two equality constraints $\hat{\boldsymbol{T}} \mathbf{1}_{M+1} = \boldsymbol{\alpha}$, and $\hat{\boldsymbol{T}}^T \mathbf{1}_{N+1} = \boldsymbol{\beta}$, respectively. The detailed derivation of this optimal $\hat{\boldsymbol{T}}_{\lambda_1,\lambda_2}$ is provided in appendix section A.1.

**Contrastive Regularization.** Incorporating temporal priors into the video alignment processes often leads to trivial solutions [41, 46]. So, following [41, 1], we utilize the Contrastive-Inverse Difference Moment (C-IDM) loss to further regularize the training. This loss is characterized by,

$$I(\boldsymbol{X}) = \sum_{i=1}^{N+1} \sum_{j=1}^{M+1} (1 - \mathcal{N}(i,j)) \gamma(i,j) \max(0, \lambda_3 - d(i,j)) + \mathcal{N}(i,j) \frac{d(i,j)}{\gamma(i,j)}, \tag{10}$$

where $\gamma(i,j) = (i-j)^2 + 1$, $d(i,j) = \|\boldsymbol{x}_i - \boldsymbol{x}_j\|$, $\mathcal{N}(i,j)$ is a neighborhood function defined as: $\mathcal{N}(i,j) = 1$, if $|i-j| \leq \delta$ and $0$ otherwise, $\delta$ is a predefined window size, $\lambda_3$ is a margin parameter. The preceding C-IDM loss is an intra-video loss. Additionally, we incorporate an inter-video contrastive loss guided by OT to further regularize the training process. Specifically, this novel loss component contrasts pairs of videos based on their similarity as quantified by the OT matrix. We find, $x_{\text{best}}(i) = \arg \max_j \hat{\boldsymbol{T}}_{\lambda_1,\lambda_2}$ and $x_{\text{worst}}(i) = \arg \min_j \hat{\boldsymbol{T}}_{\lambda_1,\lambda_2}$. Likewise, $y_{\text{best}}(j) = \arg \max_i \hat{\boldsymbol{T}}_{\lambda_1,\lambda_2}$ and $y_{\text{worst}}(j) = \arg \min_i \hat{\boldsymbol{T}}_{\lambda_1,\lambda_2}$ are calculated. Then, the best distance is computed as the average of squared differences between matched pairs, scaled by a temperature factor: best_distance $= \frac{1}{\text{temperature}} \cdot \left( \frac{1}{N} \sum_{i=1}^{N} \|\boldsymbol{x}_i - \boldsymbol{y}_{x_{\text{best}}(i)}\|^2 + \frac{1}{M} \sum_{j=1}^{M} \|\boldsymbol{y}_j - \boldsymbol{x}_{y_{\text{best}}(j)}\|^2 \right)$. Similarly, the worst distance is: worst_distance $= \frac{1}{\text{temperature}} \cdot \left( \frac{1}{N} \sum_{i=1}^{N} \|\boldsymbol{x}_i - \boldsymbol{y}_{x_{\text{worst}}(i)}\|^2 + \frac{1}{M} \sum_{j=1}^{M} \|\boldsymbol{y}_j - \boldsymbol{x}_{y_{\text{worst}}(j)}\|^2 \right)$. Finally, the inter-sequence loss is computed using the cross-entropy over the best and worst distances:

$$\text{loss\_inter} = F_{\text{cross\_entropy}} \left( \begin{bmatrix} \text{best\_distance} \\ \text{worst\_distance} \end{bmatrix}, \begin{bmatrix} 0 \\ 1 \end{bmatrix} \right). \tag{11}$$

Ideally, we want each frame embedding $\boldsymbol{x}_i$, to align highly to its best match from $\boldsymbol{Y}$. So the best distance should be as close to 0 as possible, at the same time, we maximize its distance from the unmatched frame embeddings, and the same holds true for $\boldsymbol{y}_j$s. As a result, our proposed inter-video loss (Eqn. (11)) promotes learning disentangled representations. So, the overall loss for OPEL combines the regularized OT loss (Eqn.( 9)) with the contrastive regularization terms,

$$L_{\text{OPEL}}(\boldsymbol{X},\boldsymbol{Y}) = c_1 * l_{\lambda_1,\lambda_2}^R(\boldsymbol{X},\boldsymbol{Y}) + c_2 * (I(\boldsymbol{X}) + I(\boldsymbol{Y})) + c_3 * \text{loss\_inter}. \tag{12}$$

**Clustering and Key-step Ordering.** After learning the embeddings, our goal is to localize the key-steps required for PL. We frame this problem as multi-label graph-cut segmentation [50]. The node set $V$ of the graph includes $k$ terminal nodes representing the key-steps and non-terminal nodes corresponding to the number of frames, which are derived from the embeddings produced by the embedder network. Upon constructing the graph, we apply $\alpha$-Expansion [19] to identify the minimum cost cut, utilizing the results to assign frames to $k$ labels. To deduce the sequential order of key-steps, we first compute the normalized time for each frame in a video, following [1]. Subsequently, the temporal instant for each cluster is determined by calculating the average normalized time for frames allocated to that cluster. Clusters are then sequenced in ascending order of their average time, thus outlining the sequence of key-steps of a video. Upon establishing the key-step order for all videos associated with the same task, we generate a ranked list based on the frequency at which subjects adhere to a specific sequence. The most commonly observed order is placed at the top of this list. This methodological approach allows us to discern various sequential orders of key-steps of a task.

# 4  Experiments and Results

**Datasets.** In contrast to previous research that predominantly utilized either $1^{st}$ person or $3^{rd}$ person viewpoints for PL, we incorporate datasets from both perspectives. For $3^{rd}$ person view, we utilize established benchmark datasets, namely CrossTask [11] and ProceL [3]. CrossTask features 213 hours of video footage spanning 18 primary tasks, totaling 2763 videos. ProceL includes 47.3 hours of video from 12 varied tasks, comprising 720 videos. To evaluate the effectiveness of our proposed OPEL framework, we apply it to the $1^{st}$-person EgoProceL benchmark [1], which contains 62 hours of egocentric video recordings from 130 subjects engaged in 16 tasks. Detailed information on individual datasets is provided in Table A2 of appendix.

**Evaluation.** Unless specified differently, we assess OPEL as per the current SOTA [1, 2]. We compute the framewise scores for each key-step separately and then take the mean of the scores over all the key-steps, reporting both the F1-score and the Intersection over Union (IoU). The F1-score is defined as the harmonic mean of precision and recall. Precision is calculated as the ratio of the number of frames correctly predicted as key-steps to the total number of frames labeled as key-steps. Recall is determined by the ratio of correctly predicted key-step frames to the total number of actual key-step frames. Following the methodology in [1, 2, 31, 3, 47], we employ the Hungarian algorithm [51] to derive a one-to-one mapping between the ground truth and the predictions.

Table 1: Results using EgoProceL [1] demonstrate the superior performance of OPEL. The results in bold and underline denote the highest and second-highest values in a column, respectively.

| | EgoProceL | | | | | | | | | | | |
| | CMU-MMAC [17] | | EGTEA-GAZE+[52] | | MECCANO[53] | | EPIC-Tents[54] | | PC Assembly | | PC Disassembly | |
| | F1 | IoU | F1 | IoU | F1 | IoU | F1 | IoU | F1 | IoU | F1 | IoU |
|---|---|---|---|---|---|---|---|---|---|---|---|---|
| Random | 15.7 | 5.9 | 15.3 | 4.6 | 13.4 | 5.3 | 14.1 | 6.5 | 15.1 | 7.2 | 15.3 | 7.1 |
| Uniform | 18.4 | 6.1 | 20.1 | 6.6 | 16.2 | 6.7 | 16.2 | 7.9 | 17.4 | 8.9 | 18.1 | 9.1 |
| CnC [1] | 22.7 | 11.1 | 21.7 | 9.5 | 18.1 | 7.8 | 17.2 | 8.3 | 25.1 | 12.8 | 27.0 | 14.8 |
| GPL-2D [2] | 21.8 | 11.7 | 23.6 | 14.3 | 18.0 | 8.4 | 17.4 | 8.5 | 24.0 | 12.6 | 27.4 | 15.9 |
| UG-I3D [2] | 28.4 | 15.6 | 25.3 | 14.7 | 18.3 | 8.0 | 16.8 | 8.2 | 22.0 | 11.7 | 24.2 | 13.8 |
| GPL-w BG [2] | 30.2 | 16.7 | 23.6 | 14.9 | 20.6 | 9.8 | 18.3 | 8.5 | 27.6 | 14.4 | 26.9 | 15.0 |
| GPL-w/o BG [2] | 31.7 | 17.9 | 27.1 | **16.0** | 20.7 | 10.0 | 19.8 | 9.1 | 27.5 | 15.2 | 26.7 | 15.2 |
| OPEL *(Ours)* | **36.5** | **18.8** | **29.5** | 13.2 | **39.2** | **20.2** | **20.7** | **10.6** | **33.7** | **17.9** | **32.2** | **16.9** |

**Experimental Setup.** We employ ResNet-50 (pretrained on ImageNet) as the embedder network. Inspired by [1], we train the embedder using pairs of training videos. Within these videos, we randomly select frames and optimize the proposed $L_{OPEL}$ until convergence. The feature extraction is conducted from the Conv4c layer, and we subsequently create a stack of 2 context frames along the temporal dimension. Our video frames are resized to 224×224. The aggregated features are processed through two 3D convolutional layers, followed by a 3D global max pooling layer, two fully-connected layers, and a linear projection layer that outputs embeddings of 128 dimensions. All hyper-parameters are listed in Table A1. Our code is provided as part of the supplementary material.

**Results on Egocentric View.** Table 1 presents a comparative analysis between the SOTA techniques and OPEL applied to the large scale egocentric benchmark, EgoProceL. Results from tasks within CMU-MMAC and EGTEA G. have been aggregated and presented (detailed task-wise results are given in Table A4). It is important to highlight that EgoProceL represents a contemporary dataset specifically designed for egocentric procedure learning, thereby limiting the number of applicable approaches for fair comparison. Notably, OPEL outperforms the SOTA across most tasks. This superiority underscores the efficacy of the video representation learning through OT. Specifically, we achieve 22.4% (IoU) and 26.9% (F1) average improvement compared to current SOTA.

Table 2: PL results on third-person datasets [3, 11]. P, R, and F1 represent precision, recall, F1-score.

| | ProceL [3] | | | CrossTask [11] | | |
| | P | R | F1 | P | R | F1 |
|---|---|---|---|---|---|---|
| Uniform | 12.4 | 9.4 | 10.3 | 8.7 | 9.8 | 9.0 |
| Alayrc *et al.* [34] | 12.3 | 3.7 | 5.5 | 6.8 | 3.4 | 4.5 |
| Kukleva *et al.* [32] | 11.7 | 30.2 | 16.4 | 9.8 | 35.9 | 15.3 |
| Elhamifar *et al.* [3] | 9.5 | 26.7 | 14.0 | 10.1 | **41.6** | 16.3 |
| Fried *et al.* [37] | - | - | - | - | 28.8 | - |
| Shen *et al.* [47] | 16.5 | 31.8 | 21.1 | 15.2 | 35.5 | 21.0 |
| CnC [1] | 20.7 | 22.6 | 21.6 | 22.8 | 22.5 | 22.6 |
| GPL-2D [2] | 21.7 | 23.8 | 22.7 | 24.1 | 23.6 | 23.8 |
| UG-I3D [2] | 21.3 | 23.0 | 22.1 | 23.4 | 23.0 | 23.2 |
| GPL [2] | 22.4 | 24.5 | 23.4 | 24.9 | 24.1 | 24.5 |
| STEPS [16] | 23.5 | 26.7 | 24.9 | 26.2 | 25.8 | 25.9 |
| OPEL *(Ours)* | **33.6** | **36.3** | **34.9** | **35.6** | 34.8 | **35.1** |

**Results on Third-person View.** We provide comparison between SOTA and OPEL on two distinct third-person datasets [3, 11] in Table 2. To ensure consistency in evaluation, we follow the evaluation protocol outlined in the SOTA prior arts [1, 2]. Once again, we perform better in almost all cases, with

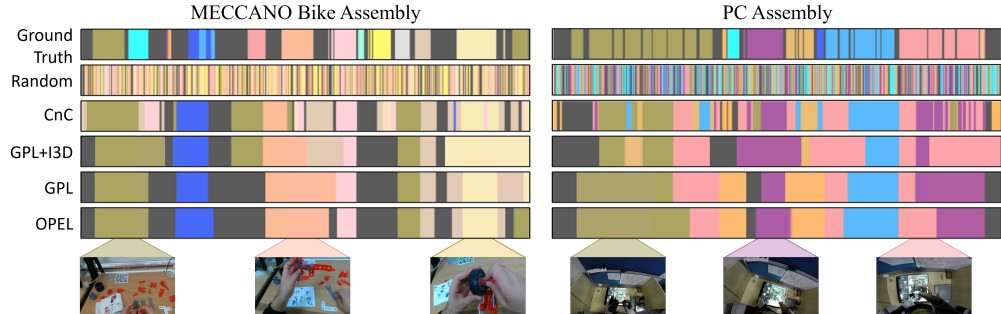

Figure 3: Qualitative results from MECCANO [53] and PC Assembly [1] tasks. Each sub-task is color-coded to represent different key-steps, while gray areas signify background elements. Notably, OPEL's performance surpasses that of the SOTA networks, attributed to its ability to handle unmatched frames through the integration of a virtual frame, thus enhancing alignment accuracy.

46.2% (F1) average enhancement over SOTA. Note, [32, 3] predominantly allocate frames to a single key-step, resulting in elevated recall rates but concomitantly diminishing precision, consequently impacting the overall F-score. Additional detailed third-person results from CMU-MMAC [17], ProceL [3] and CrossTask [11] are given in Table A3 and Table A5, respectively.

**Qualitative Results.** Fig. 3 illustrates the qualitative PL outcomes of the baselines and OPEL. Higher match with the ground truth in our case (the bottom row) depicts the usefulness of OPEL. Additionally, we depict the alignment of two sequences in Fig. 4(B), showcasing accurate alignment despite temporal variations, with correct matches indicating consistent action frame alignment and redundancy handling, affirming the reliability of our model.

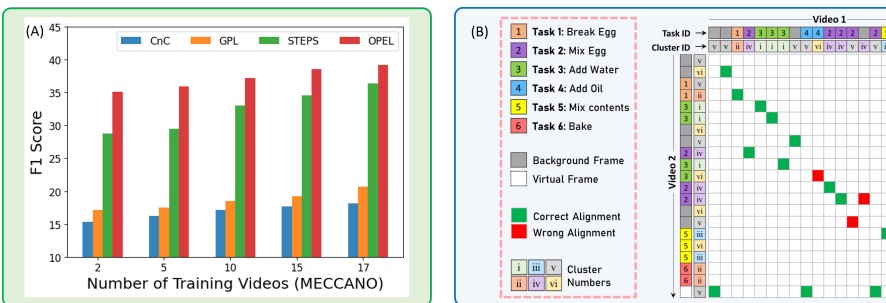

Figure 4: (A) Impact of training data quantity on encoder training. (B) Example alignment of two videos with corresponding key-step clusters from the Brownie task [17].

**Comparison with Multimodal Models.** While we only use videos for training, our results are competitive with models using multiple modalities. On the egocentric EgoProcel dataset, we perform comparably or even better (4 out of 6 datasets) compared to the multimodal SOTA model, STEPS [16], as shown in Table 3. Note, STEPS uses gaze and depth data during training, thus enhancing its results on EPIC-Tents. We also outperform [47, 34] (Table 2), which use narrations with video.

Table 3: Comparison with models with multimodal input. Note, STEPS [16] uses additional data (optical flow, gaze, depth) for training, while we use just the visual modality.

|  | CMU-MMAC | | EGTEA-GAZE+ | | MECCANO | | EPIC-Tents | | ProceL | | CrossTask | |
|---|---|---|---|---|---|---|---|---|---|---|---|---|
|  | F1 | IoU | F1 | IoU | F1 | IoU | F1 | IoU | F1 | IoU | F1 | IoU |
| STEPS [16] | 28.3 | 11.4 | **30.8** | 12.4 | 36.4 | 18.0 | **42.2** | **21.4** | 24.9 | 15.4 | 25.9 | 14.6 |
| OPEL | **36.5** | **18.8** | 29.5 | **13.2** | **39.2** | **20.2** | 20.7 | 10.6 | **34.9** | **21.3** | **35.1** | **21.5** |

## 5 Ablation Study

**Effectiveness of $L_{\text{OPEL}}$** We analyze the effectiveness of the proposed loss by replacing or combining with other SOTA losses used for PL - TCC [8], LAV [41], and CnC [1]. Overall, the proposed $L_{\text{OPEL}}$ outperforms previous approaches as shown in Table 4. This enhancement can be attributed to the flexibility in modeling sequences provided by OT. Furthermore, ablation results on all the loss components of $L_{\text{OPEL}}$ are provided in Table 5, where we show the contribution of each factor individually to analyze their effect on the overall result. Comparing row 3 with row 9, we observe, the priors jointly play a critical role; as without them (row 3), the F1 and IoU scores drop by ∼5 points.

Specifically, the optimality prior has a significant impact ($\sim$2 point), while the temporal prior affects the score by $\sim$1 point. Similar to the combined priors, the intra and inter-video contrastive losses together (row 6 vs row 9) have a significant effect ($\sim$3.5 points) on the overall performance. The individual effect of virtual frame is negligible as it only plays a role in case of excessive background frames - a scenario that is not prevalent in most datasets. Furthermore, due to the IDM structure of $M(\hat{T})$, the $\hat{T}$ and $\hat{Q}$ are similar by formulation. This results in KL($\hat{T} \parallel \hat{Q}$) to be already small. As a consequence, adding the KL divergence as a standalone loss component in the proposed pipeline has a minimal impact. Overall, while some loss components may have a smaller individual impact, they do contribute to performance improvements, even if incrementally. Therefore, our proposed approach incorporates all of them to achieve the best possible results.

Table 4: Comparison of effectiveness of $L_{\text{OPEL}}$ with other losses.

|  | CMU-MMAC [17] | | | MECCANO [53] | | | EGTEA-GAZE+ [52] | | | PC Assembly [1] | | |
|---|---|---|---|---|---|---|---|---|---|---|---|---|
|  | P | F1 | IoU | P | F1 | IoU | P | F1 | IoU | P | F1 | IoU |
| TCC + PCM [8] | 18.5 | 19.7 | 9.5 | 15.1 | 17.9 | 8.7 | 17.5 | 19.7 | 8.8 | 19.9 | 21.7 | 11.6 |
| LAV + TCC + PCM [41] | 18.8 | 19.7 | 9.0 | 13.4 | 15.6 | 7.3 | 16.4 | 18.6 | 7.5 | 21.6 | 21.1 | 10.8 |
| LAV + PCM [41] | 20.6 | 21.1 | 9.4 | 14.6 | 17.4 | 7.1 | 17.4 | 19.1 | 8.0 | 21.5 | 22.7 | 11.7 |
| TC3I + PCM (CnC) [1] | 21.6 | 22.7 | 11.1 | 15.5 | 18.1 | 7.8 | 19.6 | 21.7 | 9.5 | 25.0 | 25.1 | 12.8 |
| OT + TCC | 28.8 | 32.6 | 15.6 | 25.2 | 34.5 | 17.5 | 22.6 | 26.7 | 11.2 | 27.8 | 28.2 | 15.6 |
| OT + LAV | 30.2 | 34.7 | 16.8 | 26.7 | 36.2 | 18.8 | 23.1 | 27.8 | 12.4 | 30.2 | 30.9 | 16.8 |
| OT + TCC + LAV | 27.6 | 31.2 | 15.3 | 23.8 | 33.6 | 16.1 | 21.8 | 25.4 | 10.5 | 28.1 | 28.4 | 14.7 |
| OPEL *(Ours)* | **32.8** | **36.5** | **18.8** | **28.9** | **39.2** | **20.2** | **24.3** | **29.5** | **13.2** | **32.5** | **33.7** | **17.9** |

Table 5: Analysis of the impact of each term in $L_{\text{OPEL}}$ on the overall performance.

| Intra-Video | Inter-Video | KL Divergence | Temporal Prior | Optimality Prior | Virtual Frame | MECCANO [53] | | CMU-MMAC [17] | |
|---|---|---|---|---|---|---|---|---|---|
|  |  |  |  |  |  | F1 | IoU | F1 | IoU |
| ✓ |  |  |  |  |  | 34.1 | 14.2 | 30.5 | 12.9 |
|  | ✓ |  |  |  |  | 33.3 | 13.5 | 29.6 | 12.3 |
| ✓ | ✓ |  |  |  |  | 34.6 | 14.9 | 31.3 | 13.7 |
| ✓ | ✓ | ✓ | ✓ |  |  | 36.1 | 18.4 | 33.8 | 16.4 |
| ✓ | ✓ | ✓ |  | ✓ |  | 38.6 | 19.6 | 36.1 | 18.2 |
|  |  | ✓ | ✓ | ✓ | ✓ | 35.8 | 16.1 | 32.6 | 14.4 |
| ✓ | ✓ | ✓ |  |  | ✓ | 37.0 | 18.3 | 34.1 | 16.5 |
| ✓ | ✓ |  | ✓ | ✓ | ✓ | 38.1 | 19.1 | 35.2 | 17.3 |
| ✓ | ✓ | ✓ | ✓ | ✓ | ✓ | **39.2** | **20.2** | **36.5** | **18.8** |

Table 6: Analysis of different clustering algorithms.

|  | CMU-MMAC | | EGTEA-GAZE+ | | MECCANO | | EPIC-Tents | | ProceL | | CrossTask | |
|---|---|---|---|---|---|---|---|---|---|---|---|---|
|  | F1 | IoU | F1 | IoU | F1 | IoU | F1 | IoU | F1 | IoU | F1 | IoU |
| Random | 15.7 | 5.9 | 15.3 | 4.6 | 13.4 | 5.3 | 14.1 | 6.5 | 15.1 | 7.2 | 15.3 | 7.1 |
| OT + K-means | 34.2 | 13.5 | 23.9 | 8.8 | 31.8 | 19.6 | 16.2 | 7.9 | 24.8 | 12.5 | 27.4 | 14.4 |
| OT + SS | 34.8 | 13.2 | 23.7 | 8.7 | 31.6 | 19.5 | 17.2 | 8.3 | 25.1 | 12.8 | 28.0 | 14.8 |
| OPEL | **36.5** | **18.8** | **29.5** | **13.2** | **39.2** | **20.2** | **20.7** | **10.6** | **33.7** | **17.9** | **32.2** | **16.9** |

**Choice of clustering algorithm.** We replace the proposed clustering approach with K-means and subset selection (SS). The results in Table 6 show that OPEL performs the best, highlighting the effectiveness of OT with graphcut segmentation.

**Number of key-steps.** In Table 7, we present the results of OPEL alongside baseline models, with varying $k$. Note, we obtain best results with $k$=7, and the performance drops sharply as $k$ goes from 7 to 10 or higher. This observation is consistent with all the other SOTA methods on the same datasets [1, 2, 16]. We hypothesize that $k$=7 works best as it is the optimal number of clusters considering the average number of distinct key-steps (subtasks) of the datasets. For example, for PC Disassembly, although the ground-truth (GT) number of steps is 9, 3 steps are quite similar (remove hard disk, remove motherboard, remove RAM), effectively making them quite close in the feature space. This results in $k$=7 being a better estimation of the cluster number with distinct steps. Note,

Table 7: Results obtained for different $k$.

| $k$ | PC Assembly | | | PC Disassembly | | |
|---|---|---|---|---|---|---|
|  | R | F1 | IoU | R | F1 | IoU |
| 7 | **35.0** | **33.7** | **18.0** | **35.4** | **32.2** | **16.7** |
| 10 | 27.8 | 24.3 | 12.1 | 28.5 | 24.8 | 10.5 |
| 12 | 25.2 | 24.1 | 11.8 | 26.7 | 24.2 | 9.7 |
| 15 | 27.6 | 25.8 | 12.2 | 25.2 | 23.6 | 9.1 |

this demarcation of subtasks (hence, number of clusters) is subjective and varies from dataset to dataset as well as from task to task; as some may consider semantically similar tasks (e.g. pouring oil vs water) to be one subtask, while others may consider it different. As $k$ becomes larger than the actual distinctive number of clusters, each subtask gets split into multiple clusters with very similar embeddings, which upon comparison with GT leads to inferior results.

**Impact of Training data Quantity.** Fig. 4(A) presents the results from varying the number of training videos on MECCANO, aiming to evaluate OPEL's performance with respect to video count. We consistently outperform other SOTA methods. Overall, the performance improves with more training data, however, even with just few (2-5) videos of a task, we reach the upper-limit of other methods using full dataset, as shown in Fig. 4(A). Additional ablation results including choice of distribution as priors and hyperparameters $\lambda_1, \lambda_2$ are provided in appendix A.8.

**Comparison with AS methods.** PL and action segmentation (AS) are related but not the same. PL, when applied to a set of instructional videos depicting the same task, involves two primary steps: (i) assigning each video frame to one of the $k$ key-steps (including background elements), and (ii) determining the logical sequence of these key-steps necessary to complete the task. As illustrated in Fig. 1, PL addresses multiple videos of a given task, enabling the identification of repetitive key-steps across these videos [1, 2]. In contrast, AS [4] focuses on a single video, thereby lacking the ability to discern repetitive key-steps across different videos.

Despite the differences between PL and AS, we compare our approach against existing SOTA unsupervised AS models and present the results in Table 8. Our model demonstrates a significant performance improvement compared to these works. In [3, 56], authors report a high recall score for CrossTask as it assigns majority of the frames to a single key-step - a phenomenon also reported by [2]. While achieving high recall is important for ensuring that most positive instances are correctly identified, it can result in a greater number of false positives, which in turn lowers precision and leads to undesirable

Table 8: Comparison with SOTA unsupervised AS methods. Note '-' denotes that the authors have not provided any data on those metrics.

| AS benchmark | ProceL [3] | | | CrossTask [11] | | |
|---|---|---|---|---|---|---|
| | P | R | F1 | P | R | F1 |
| JointSeqFL [31] | - | - | 29.8 | - | - | - |
| Elhamifar *et al.* [3] | 9.5 | 26.7 | 14.0 | 10.1 | 41.6 | 16.3 |
| Fried *et al.* [37] | - | - | - | - | 28.8 | - |
| Shen *et al.* [47] | 16.5 | 31.8 | 21.1 | 15.2 | 35.5 | 21.0 |
| Dvornik *et al.* [55] | - | - | - | - | - | 25.3 |
| StepFormer [56] | 18.3 | 28.1 | 21.9 | 22.1 | **42** | 28.3 |
| OPEL *(Ours)* | **33.6** | **36.3** | **34.9** | **35.6** | 34.8 | **35.1** |

results. Therefore, it is crucial to balance recall with precision to develop an effective model. This balance is reflected in the superior performance of our model, as evidenced by the F1-score results across various benchmarks. Note in the Table 8, our approach is compared with SOTA unsupervised AS methods for only third-person datasets, as these do not report any result on egocentric datasets.

## 6 Conclusion

In this study, we have introduced a novel approach for procedure learning leveraging optimal transport, enhanced by temporal and distributional regularizations to improve the alignment of key-steps across multiple video instances. Our method addresses inherent limitations in current SOTA techniques that primarily rely on frame-to-frame mappings and assumptions of monotonic alignment, which do not optimally utilize temporal information. We observe an improvement of 22.4% in IoU and 26.9% in F1 scores on the EgoProceL dataset, outperforming the current state-of-the-art methods. Similarly, in third-person video benchmarks, such as ProCeL and CrossTask, our framework achieves an average F1 score enhancement of 46.2% over existing methods. These advancements underscore the potential of OT guided learning in handling complex video procedure learning tasks. A limitation of the proposed OPEL framework is the assumption that subjects utilize similar objects for identical key-steps, which may introduce inaccuracies when dissimilar objects are employed in the execution of these steps. Future work will focus on exploring the integration of additional contextual and semantic features within the OT framework to further refine the procedure learning process. Moreover, extending this framework to other domains of video understanding could provide valuable insights into the general applicability of optimal transport in video analysis tasks.

## Acknowledgments

This work was supported in part by the Center for Co-design of Cognitive Systems (CoCoSys), one of the seven centers in JUMP 2.0, a Semiconductor Research Corporation (SRC) program sponsored by DARPA, by the SRC, the National Science Foundation, Intel Corporation, the DoD Vannevar Bush Fellowship, and by the U.S. Army Research Laboratory.

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

## A Appendix

### A.1 Derivation of the Optimal Transport Matrix $(\hat{T}_{\lambda_1,\lambda_2})$

In this section, we obtain the optimal transport matrix $\hat{T}_{\lambda_1,\lambda_2}$ that optimizes the OT Eqn. 9. Note, all the notations used here are same as section 3 of the main manuscript. We start with Eqn. 9 of section 3-

$$l^R_{\lambda_1,\lambda_2}(\boldsymbol{X},\boldsymbol{Y}) := \langle \hat{T}_{\lambda_1,\lambda_2}, \boldsymbol{D} \rangle, \text{ s.t. } \hat{T}_{\lambda_1,\lambda_2} = \arg\min_{\hat{T}\in U(\boldsymbol{\alpha},\boldsymbol{\beta})} \langle \hat{T}_{\lambda_1,\lambda_2}, \boldsymbol{D} \rangle - \lambda_1 M(\hat{T}) + \lambda_2 \text{KL}(\hat{T} \parallel \hat{Q}),$$

From the duality theory, we know, for each pair $\xi_1, \xi_2$ in Equation 8, a corresponding pair $\lambda_1 > 0$, $\lambda_2 > 0$ exists, such that $l^R_{\xi_1,\xi_2}(\boldsymbol{X},\boldsymbol{Y}) = l^R_{\lambda_1,\lambda_2}(\boldsymbol{X},\boldsymbol{Y})$ for the pair $(\boldsymbol{X},\boldsymbol{Y})$.

$\hat{T}_{\lambda_1,\lambda_2}$ is the optimal transport matrix, so it optimizes-

$$\min_{\hat{T}\in\mathbb{R}^{N+1\times M+1}_+} \langle \hat{T}, \boldsymbol{D} \rangle - \lambda_1 M(\hat{T}) + \lambda_2 \text{KL}(\hat{T} \parallel \hat{Q}) \quad \text{subject to} \quad \hat{T}\mathbf{1}_{M+1} = \boldsymbol{\alpha}, \hat{T}^\top \mathbf{1}_{N+1} = \boldsymbol{\beta}, \tag{A1}$$

Given that both the objective function and the feasible set defined in Equation A1 are convex, the existence and uniqueness of the optimal transport matrix $\hat{T}_{\lambda_1,\lambda_2}$ are guaranteed. To derive this optimal matrix, the analysis begins by taking the Lagrangian of Equation A1 as-

$$L(\hat{T},\boldsymbol{\mu},\boldsymbol{\nu}) = \sum_{i=1}^{N+1}\sum_{j=1}^{M+1}\left( d_{ij}t_{ij} - \lambda_1 t_{ij}\left( \frac{1}{\left(\frac{i}{N+1}-\frac{j}{M+1}\right)^2+1} + \frac{1}{\frac{1}{2}d_m+1} \right) + \lambda_2 t_{ij}\log\frac{t_{ij}}{q_{ij}} \right)$$
$$+ \boldsymbol{\mu}^T(\hat{T}\mathbf{1}_{M+1}-\boldsymbol{\alpha}) + \boldsymbol{\nu}^T(\hat{T}^\top\mathbf{1}_{N+1}-\boldsymbol{\beta}), \tag{A2}$$

where $d_m = \left(\frac{i-i_o}{N+1}\right)^2 + \left(\frac{j-j_o}{M+1}\right)^2$, $i_o$ and $j_o$ have their same meaning as the main paper, i.e. they correspond to the optimal assignment locations $(i,j_o)$ and $(i_o,j)$ as provided by the transport matrix, and $\boldsymbol{\mu}$ and $\boldsymbol{\nu}$ are the dual variables for the two equality constraints $\hat{T}\mathbf{1}_{M+1} = \boldsymbol{\alpha}$, and $\hat{T}^T\mathbf{1}_{N+1} = \boldsymbol{\beta}$, respectively. Taking the derivative of $L(\hat{T},\boldsymbol{\mu},\boldsymbol{\nu})$ w.r.t. $t_{ij}$ yields-

$$\frac{\partial L(\hat{T},\boldsymbol{\mu},\boldsymbol{\nu})}{\partial t_{ij}} = d_{ij} - \lambda_1\left( \frac{1}{\left(\frac{i}{N+1}-\frac{j}{M+1}\right)^2+1} + \frac{1}{\frac{1}{2}\left(\left(\frac{i-i_o}{N+1}\right)^2+\left(\frac{j-j_o}{M+1}\right)^2\right)+1} \right)$$
$$+ \lambda_2\log\frac{t_{ij}}{q_{ij}} + \lambda_2 + \mu_i + \nu_j. \tag{A3}$$

Let, $s^{\lambda_1}_{ij} = \lambda_1\left( \frac{1}{\left(\frac{i}{N+1}-\frac{j}{M+1}\right)^2+1} + \frac{1}{\frac{1}{2}\left(\left(\frac{i-i_o}{N+1}\right)^2+\left(\frac{j-j_o}{M+1}\right)^2\right)+1} \right).$

Setting $L(\hat{T},\boldsymbol{\mu},\boldsymbol{\nu}) = 0$, we get-

$$t_{ij} = q_{ij}e^{-\frac{1}{2}-\frac{\mu_i}{\lambda_2}}e^{\frac{1}{\lambda_2}(s^{\lambda_1}_{ij}-d_{ij})}e^{-\frac{1}{2}-\frac{\nu_j}{\lambda_2}}. \tag{A4}$$

Let, $\boldsymbol{K} = [q_{ij}e^{\frac{1}{\lambda_2}(s^{\lambda_1}_{ij}-d_{ij})}]_{ij}$, then we get,

$$t_{ij} = e^{-\frac{1}{2}-\frac{\mu_i}{\lambda_2}}\boldsymbol{K}_{ij}e^{-\frac{1}{2}-\frac{\nu_j}{\lambda_2}},$$

Thus,

$$\hat{T}_{\lambda_1,\lambda_2} = e^{diag(-\frac{1}{2}-\frac{\mu}{\lambda_2})} K e^{diag(-\frac{1}{2}-\frac{\nu}{\lambda_2})}. \tag{A5}$$

Every element of the matrix $K$ is strictly positive, as we take the element-wise exponential to obtain each $K_{ij}$ and $q_{ij} > 0$. As per the Sinkhorn's theorem (Theorem A), there exist diagonal matrices $diag(\kappa_1)$ and $diag(\kappa_2)$ with strictly positive diagonal elements, such that $diag(\kappa_1) K diag(\kappa_2)$ is a member of the set $U(\alpha, \beta)$, with $\kappa_1 \in \mathbb{R}^{N+1}$ and $\kappa_2 \in \mathbb{R}^{M+1}$. This product matrix is unique, and the diagonal matrices are also uniquely determined, up to a scalar factor.

**Theorem A** [57, 58]: For any $(N + 1) \times (M + 1)$ matrix $A$ with all positive elements, diagonal matrices $B_1$ and $B_2$ exist such that $B_1 A B_2$ belongs to $U(\alpha, \beta)$. Both $B_1$ and $B_2$ possess strictly positive diagonal elements and are unique up to a positive scalar factor.

The optimal $\hat{T}_{\lambda_1,\lambda_2}$ of Equation (A5) in $U(\alpha, \beta)$ mirrors the form of $diag(\kappa_1) K diag(\kappa_2)$, thereby constituting the unique matrix in $U(\alpha, \beta)$ that represents a rescaled version of $K$. We efficiently compute the scaling vectors $\kappa_1$ and $\kappa_2$, also unique up to a scaling factor, using the Sinkhorn-Knopp iterative matrix scaling algorithm-

$$\kappa_1 \leftarrow \frac{\alpha}{K\kappa_2},$$

$$\kappa_2 \leftarrow \frac{\beta}{K^T\kappa_1}.$$

In this paper, only 20 iterations are used, as a limited number of iterations has been shown to effectively converge in previous studies [48].

## A.2 Hyper-parameter Settings

Table A1 lists the hyper-parameters used for OPEL.

Table A1: Hyper-parameter settings for OPEL.

| Hyper-parameter | Value |
|---|---|
| No. of key-steps ($k$) | 7 |
| No. of sampled frames ($N, M$) | 32 |
| No. of epochs | 10000 |
| Batch Size | 2 |
| Learning Rate | $10^{-4}$ |
| Weight Decay | $10^{-5}$ |
| Window size ($\delta$) | 15 |
| Laplace scale parameter ($b$) | 3.0 (MECCANO, EPIC-Tents PC Assembly) |
| Laplace scale parameter ($b$) | 2.0 (for all other datasets) |
| Temperature | 0.5 |
| $\lambda_1$ | $\frac{1}{N+M}$ |
| $\lambda_2$ | $\frac{0.1*N*M}{4.0}$ |
| Margin ($\lambda_3$) | 2.0 |
| Threshold for virtual frame ($\zeta$) | $\frac{2*5}{N+M}$ |
| No. of context frames | 2 |
| Context stride | 15 |
| Embedding Dimension | 128 |
| Optimizer | Adam |
| $c_1$ | $\frac{1}{N*M}$ |
| $c_2$ | 0.5 |
| Coefficient for loss_inter ($c_3$) | 0.0001 |
| Maximum Sinkhorn Iterations | 20 |

## A.3 Compute Resources for Experiments

For our experiment, we require adequate computing resources to effectively train our models. We utilize a single Nvidia A40 GPU, but its full RAM is not required. The GPU memory is

dependent on batch size (bs). For a bs of 2, a GPU equipped with approximately 12GB of memory is sufficient for our purposes. Training time depends on dataset size and number of epochs (we used 10000). The above configuration allows us to process a dataset consisting of 15-20 videos (e.g. PC assembly/MECCANO domain) in around 12 hours. With these computing resources in place, we conducted our experiments effectively, ensuring optimal performance and reliable outcomes.

## A.4 Detailed Statistics of Dataset

In Table A2, we provide the statistical analysis for each of the 16 tasks within the EgoProceL dataset [1]. $N$ denotes the total number of videos, while $K$ represents the number of key-steps for each task. $u_n$ signifies the count of unique key-steps, and $g_n$ denotes the number of annotated key-steps for the $n^{th}$ video. Following the approach outlined in reference [31], we compute the subsequent metrics:

*Foreground Ratio:* This measure indicates the proportion of the total duration occupied by key-steps in relation to the overall video duration. It assists in gauging the prevalence of background actions within a task. The foreground ratio is inversely correlated with the presence of background activity and is determined as:

$$F = \frac{\sum_{n=1}^{N} \frac{t_k^n}{t_v^n}}{N} \tag{A6}$$

Here, $t_k^n$ and $t_v^n$ represent the duration of key-steps and the video for the $n^{th}$ instance, respectively. The foreground ratio F ranges from 0 to 1. The higher the value of F indicates minimal background actions.

Table A2: Statistics of the EgoProceL dataset across different tasks.

| Task | Videos Count | Key-steps Count | Foreground Ratio | Missing Key-steps | Repeated Key-steps |
|---|---|---|---|---|---|
| PC Assembly [1] | 14 | 9 | 0.79 | 0.02 | 0.65 |
| PC Disassembly [1] | 15 | 9 | 0.72 | 0.00 | 0.60 |
| MECCANO (Toy Bike Assembly) [53] | 20 | 17 | 0.50 | 0.06 | 0.32 |
| Epic-Tents (Tent Assembly) [54] | 29 | 12 | 0.63 | 0.14 | 0.73 |
| CMU-MMAC [17] | | | | | |
| Brownie | 34 | 9 | 0.44 | 0.19 | 0.26 |
| Eggs | 33 | 8 | 0.26 | 0.05 | 0.26 |
| Pepperoni Pizza | 33 | 5 | 0.53 | 0.00 | 0.26 |
| Salad | 34 | 9 | 0.32 | 0.30 | 0.14 |
| Sandwich | 31 | 4 | 0.25 | 0.03 | 0.37 |
| EGTEAGAZE+ [52] | | | | | |
| Bacon and Eggs | 16 | 11 | 0.15 | 0.22 | 0.51 |
| Cheese Burger | 10 | 10 | 0.22 | 0.22 | 0.65 |
| Continental Breakfast | 12 | 10 | 0.23 | 0.20 | 0.36 |
| Greek Salad | 10 | 4 | 0.25 | 0.18 | 0.77 |
| Pasta Salad | 19 | 8 | 0.25 | 0.19 | 0.86 |
| Hot Box Pizza | 6 | 8 | 0.31 | 0.13 | 0.62 |
| Turkey Sandwich | 13 | 6 | 0.21 | 0.01 | 0.52 |

*Missing Key-steps:* This metric quantifies the number of omitted key-steps in each video. It is defined as:

$$M = 1 - \frac{\sum_{n=1}^{N} u_n}{KN} \tag{A7}$$

The range of M is between 0 and 1. It aids in assessing the feasibility of completing a task despite certain steps being skipped.

*Repeated Key-steps:* This metric assesses the occurrence of repeated key-steps across multiple videos. It is expressed as:

$$R = 1 - \frac{\sum_{n=1}^{N} u_n}{\sum_{n=1}^{N} g_n} \tag{A8}$$

The range of R varies between 0 and 1. Higher values of R indicate a greater recurrence of key-steps across videos. OPEL considers these repetitions, to demonstrate better performance.

## A.5 Third-Person Video Perspective

In this comparison, we assess the outcomes of training OPEL on diverse perspectives from CMU-MMAC [17]. Table A3 depicts the F1-Score and IoU scores per frame for exocentric views. We conducted our experiments on exocentric videos and achieved promising results. Through rigorous testing and analysis, our model demonstrated strong performance when trained and evaluated on this particular perspective. The obtained outcomes not only validate the effectiveness of our approach but also underscore its applicability to real-world scenarios involving both egocentric and exocentric video data.

Table A3: Third-person view results using diverse perspectives from CMU-MMAC [17]. Our findings demonstrate improved outcomes utilizing OT on egocentric as well as third-person videos, emphasizing their efficacy. P, R, and F denote precision, recall, and F-score, respectively.

| View | P | R | F1 | IoU |
|---|---|---|---|---|
| TP (Top) | 29.0 | 42.0 | 34.0 | 17.5 |
| TP (Back) | 30.7 | 43.9 | 35.9 | 19.6 |
| TP (LHS) | 38.3 | 52.7 | 44.0 | 24.3 |
| TP (RHS) | 31.8 | 42.8 | 36.2 | 18.4 |

## A.6 Quantitative results of OPEL on different subtasks across both ego- and exo-datasets

We report the results for all subtasks within the egocentric datasets in Table A4 such as CMU-MMAC [17] and EGTEA-GAZE+ [52], as well as for various third-person exocentric videos in datasets like ProceL [3] and CrossTask [11] in Table A5. Our analysis encompasses comprehensive evaluations across these diverse datasets, providing insights into the performance of our model across different perspectives and scenarios. These results offer a holistic understanding of the capabilities and effectiveness of our approach in handling varied video types and tasks, contributing significantly to the advancement of research in procedure learning and related domains.

Table A4: Results on individual subtasks of egocentric datasets.

(a) EGTEA-GAZE+[52]

| Bacon Eggs | | Cheeseburger | | Breakfast | | Greek Salad | | Pasta Salad | | Pizza | | Turkey | |
|---|---|---|---|---|---|---|---|---|---|---|---|---|---|
| F1 | IoU | F1 | IoU | F1 | IoU | F1 | IoU | F1 | IoU | F1 | IoU | F1 | IoU |
| 26.7 | 10.7 | 33.6 | 14.3 | 31.4 | 14.1 | 33.5 | 17.7 | 26.1 | 10.7 | 31.7 | 14.4 | 23.6 | 10.2 |

(b) CMU-MMAC [17]

| Brownie | | Eggs | | Pizza | | Salad | | Sandwich | |
|---|---|---|---|---|---|---|---|---|---|
| F1 | IoU | F1 | IoU | F1 | IoU | F1 | IoU | F1 | IoU |
| 34.0 | 17.1 | 28.9 | 13.3 | 37.1 | 20.6 | 41.6 | 22.1 | 41.0 | 20.9 |

Table A5: Results on individual subtasks of Third-person exocentric datasets.

(a) ProceL [3]

| Clarinet | | PB&J Sandwich | | Salmon | | Jump Car | | Toilet | | Tire Change | |
| --- | --- | --- | --- | --- | --- | --- | --- | --- | --- | --- | --- |
| F1 | IoU | F1 | IoU | F1 | IoU | F1 | IoU | F1 | IoU | F1 | IoU |
| 34.5 | 21.0 | 36.6 | 22.7 | 37.3 | 23.2 | 31.3 | 18.7 | 31.6 | 18.8 | 35.5 | 21.7 |

| Tie-Tie | | Coffee | | iPhone Battery | | Repot Plant | | Chromescast | | CPR | |
| --- | --- | --- | --- | --- | --- | --- | --- | --- | --- | --- | --- |
| F1 | IoU | F1 | IoU | F1 | IoU | F1 | IoU | F1 | IoU | F1 | IoU |
| 38.3 | 23.9 | 32.6 | 19.7 | 34.4 | 20.8 | 34.4 | 20.9 | 35.8 | 22.0 | 36.7 | 22.7 |

(b) CrossTask [11]

| 40567 | | 16815 | | 23521 | | 44047 | | 44789 | | 77721 | |
| --- | --- | --- | --- | --- | --- | --- | --- | --- | --- | --- | --- |
| F1 | IoU | F1 | IoU | F1 | IoU | F1 | IoU | F1 | IoU | F1 | IoU |
| 35.0 | 21.8 | 37.4 | 23.1 | 34.4 | 20.9 | 35.1 | 21.5 | 32.6 | 19.7 | 38.7 | 24.3 |

| 87706 | | 71781 | | 94276 | | 53193 | | 76400 | | 91515 | |
| --- | --- | --- | --- | --- | --- | --- | --- | --- | --- | --- | --- |
| F1 | IoU | F1 | IoU | F1 | IoU | F1 | IoU | F1 | IoU | F1 | IoU |
| 33.4 | 20.2 | 34.6 | 21.1 | 37.5 | 23.3 | 35.5 | 21.8 | 37.0 | 22.9 | 34.2 | 20.8 |

| 59684 | | 95603 | | 105253 | | 105222 | | 109972 | | 113766 | |
| --- | --- | --- | --- | --- | --- | --- | --- | --- | --- | --- | --- |
| F1 | IoU | F1 | IoU | F1 | IoU | F1 | IoU | F1 | IoU | F1 | IoU |
| 35.6 | 21.9 | 30.9 | 18.4 | 34.1 | 20.7 | 35.2 | 21.6 | 35.7 | 22.0 | 34.9 | 21.3 |

## A.7 Additional Applications

Learning from multiple videos of the same task opens up numerous potential applications. Firstly, in monitoring procedures, a system trained to recognize key steps can identify deviations or variations when a new person performs the task. Secondly, in guidance systems, such a system can detect the current step and suggest the next steps needed for task completion. Thirdly, in automated systems, this framework enables robotic systems to autonomously learn key steps through observation, allowing them to perform tasks independently in future instances without human assistance.

In terms of cross-modal transfer within videos, the capability to align related videos without supervision allows for the transfer of annotations or other modalities from one video to another. For example, text annotations can be applied to an entire dataset of related videos by labeling just one. Additionally, temporal modalities like sound can be transferred between videos; for instance, the sound of pouring liquids can be transferred purely based on visual cues.

Moreover, fine-grained retrieval within videos can be achieved by using nearest neighbors, enabling the retrieval of specific frames that depict various scenarios. Anomaly detection is possible by observing deviations in the video trajectories within the embedding space, helping to identify unusual activities. This ensures the proper sequence of tasks, such as jacking up a car before accessing the wheel during a tire change.

## A.8 Additional Ablation Studies

### A.8.1 Distribution for Optimality and Temporal Priors

Instead of using the Laplace distribution, we also tested a Gaussian mean distribution as described in Eqn. A9, with mean $\mu$ and variance $\sigma$ to accommodate temporal variations. Additionally, we evaluated a Uniform distribution as described in Eqn. A10. We conducted ablation experiments with various hyperparameters and summarize the best performances in Table A6. However, the Laplace distribution consistently outperformed these alternatives, leading us to adopt it for our experiments.

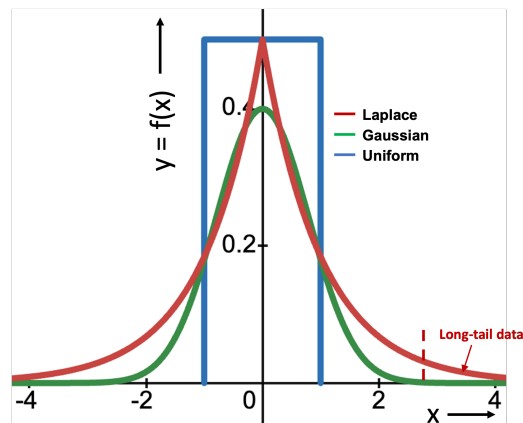

Figure A1: Importance of choosing Laplace distribution as a prior.

$$\boldsymbol{Q}(i,j) = \mathcal{N}(x; \mu, \sigma^2) = \frac{1}{\sqrt{2\pi\sigma^2}} \exp\left(-\frac{(x-\mu)^2}{2\sigma^2}\right) \tag{A9}$$

$$\boldsymbol{Q}(i,j) = f(x; a, b) = \begin{cases} \frac{1}{b-a} & \text{if } a \leq x \leq b, \\ 0 & \text{otherwise.} \end{cases} \tag{A10}$$

Table A6: Ablation on the choice of distribution function for optimality and temporal priors

| Distribution | EgoProceL | | | | | | | |
| | CMU-MMAC | | MECCANO | | PC Assembly | | PC Disassembly | |
| | F1 | IoU | F1 | IoU | F1 | IoU | F1 | IoU |
|---|---|---|---|---|---|---|---|---|
| Uniform | 31.3 | 15.2 | 28.9 | 13.8 | 26.3 | 13.5 | 27.4 | 14.2 |
| Gaussian | 35.1 | 18.3 | 33.8 | 17.3 | 29.0 | 15.3 | 30.1 | 16.5 |
| Laplace | 36.5 | 18.8 | 39.2 | 20.2 | 33.7 | 17.9 | 32.2 | 16.9 |

To analyze the superior performance of Laplace as a prior, we plot the distributions in Fig. A1. Note, we use the same distribution for optimality as well as temporal priors.

For the optimality prior, the x-axis is the difference of frames in the feature space (1-d representation for illustration purposes), and the y-axis denotes the corresponding probability of alignment. We want the point representing the most likely alignment (as per $\hat{\boldsymbol{T}}$) to have the highest likelihood, while the assignment probability should exponentially decay further away. The graph clearly shows that the Laplace distribution captures this behavior suitably compared to Uniform and Gaussian. Similarly, for the temporal prior, the x-axis denotes the temporal distance between the frames, and the y-axis denotes the corresponding probability of alignment. The graph shows that Laplace distribution facilitates alignment of the frames when they are temporally aligned (close to center), and its long tail distribution enables better correlation of non-monotonic frames compared to Gaussian or Uniform. As a result, as shown in Fig. A1, even at far away locations from the center (temporally distant frames), alignment is possible if indeed these frames have a high match feature-wise. In this case, the Laplace temporal prior provides non-zero probability to that far away frame (due to its long tail) unlike other distributions and the optimality prior gives a large score (due to feature match), resulting in improved handling of non-monotonicity.

### A.8.2 Hyperparameters $\lambda_1$ and $\lambda_2$

We present additional experiments concerning the hyperparameters used in our model, specifically $\lambda_1$ and $\lambda_2$ in Eqn. 9. We assess their impact on the EgoProceL dataset. According to Table A7, we find

that setting $\lambda_1 = \frac{1}{(N+M)}$ and $\lambda_2 = \frac{0.1*N*M}{4.0}$ achieves the best overall performance, where $N$ and $M$ denotes the number of sampled frames. Therefore, we adopt these values for the experiments reported in our paper. Additionally, the results indicate that our approach maintains similar accuracy across different combinations of $\lambda_1$ and $\lambda_2$, demonstrating robustness to the choice of hyperparameters.

Table A7: Ablation study for hyperparameters $\lambda_1$ and $\lambda_2$

| Hyperparameter | Value | EgoProceL | | | | | | | |
| | | CMU-MMAC | | MECCANO | | PC Assembly | | PC Disassembly | |
| | | F1 | IoU | F1 | IoU | F1 | IoU | F1 | IoU |
| | $\frac{0.2}{(N+M)}$ | 35.2 | 17.7 | 38.1 | 18.8 | 32.7 | 17.1 | 31.6 | 15.7 |
| $\lambda_1$ | $\frac{1}{(N+M)}$ | **36.5** | **18.8** | **39.2** | **20.2** | **33.7** | **17.9** | **32.2** | **16.9** |
| | $\frac{5}{(N+M)}$ | 35.8 | 18.1 | 38.6 | 19.3 | 32.5 | 16.7 | 31.3 | 15.6 |
| | $\frac{0.02*N*M}{4.0}$ | 34.6 | 16.9 | 36.9 | 17.8 | 31.1 | 15.4 | 29.8 | 14.2 |
| $\lambda_2$ | $\frac{0.1*N*M}{4.0}$ | **36.5** | **18.8** | **39.2** | **20.2** | **33.7** | **17.9** | **32.2** | **16.9** |
| | $\frac{0.5*N*M}{4.0}$ | 35.3 | 17.6 | 37.6 | 18.9 | 32.5 | 16.7 | 30.8 | 15.6 |

