# OpenReview forum: "OPEL: Optimal Transport Guided ProcedurE Learning"
_NeurIPS.cc/2024/Conference — NeurIPS 2024 poster_

### Official Review · Reviewer_3Xtn · 2024-07-12

**Soundness:** 3
**Presentation:** 3
**Contribution:** 3
**Rating:** 5
**Confidence:** 4

**Summary:**

The paper presents OPEL, a novel framework for procedure learning from videos that leverages optimal transport (OT) to align key steps across different video instances. OPEL treats video frames as samples from an unknown distribution and formulates the distance calculation between them as an optimal transport problem, allowing for more flexible alignment compared to frame-to-frame mappings. The authors introduce two regularization terms to improve the OT formulation and experiments significantly outperforms state-of-the-art methods on benchmark datasets.

**Strengths:**

1. This paper proposes a novel optimal transport-based procedure learning framework that aligns frames with similar semantics together in an embedding space.
2. The paper enhances the OT formulation with two regularization terms that address temporal and semantic relationships, contributing to better alignment and learning.
3. OPEL demonstrates effective performance improvements over previous state-of-the-art methods on benchmark datasets.

**Weaknesses:**

1. I believe the authors can compare with recent action segmentation methods to further strengthen the experiments.
2. I believe the authors can also compare with methods that use temporal alignment methods like dynamic time warping to strengthen their contributions.

**Questions:**

I do not have specific questions on the model design. I only have some experiment suggestions which are the same as I listed in "weaknesses".

---

> ### Author Rebuttal · Authors · 2024-08-02
>
> Thanks for your encouraging and insightful feedback. Please find the answers to your specific comments:
>
> **Comparison with AS methods:** Procedure learning (PL) and action segmentation (AS) are related but not the same. PL, when applied to a set of instructional videos depicting the same task, involves two primary steps: (i) assigning each video frame to one of  the *k* key-steps (including background elements), and (ii) determining the logical sequence of these key-steps necessary to complete the task. As illustrated in Fig. 1 of the main paper, PL addresses multiple videos of a given task, enabling the identification of repetitive key-steps across these videos [i, j]. In contrast, AS [k] focuses on a single video, thereby lacking the ability to discern repetitive key-steps across different videos.
>
> Despite the differences between PL and AS, as per the reviewer's suggestion, we compare our approach against existing SOTA unsupervised AS models [a-f] and present the results in **Table X** below. Our model demonstrates a significant performance improvement compared to these works. In [b, f], authors report a high recall score for CrossTask as it assigns majority of the frames to a single key-step - a phenomenon also reported by [j]. While achieving *high recall* is important for ensuring that most positive instances are correctly identified, it can result in a greater number of *false positives*, which in turn lowers precision and leads to undesirable results. Therefore, it is crucial to balance recall with precision to develop an effective model. This balance is reflected in the superior performance of our model, as evidenced by the F1-score results across various benchmarks. Note in the Table X, our approach is compared with SOTA **Unsupervised AS** methods for only third-person datasets, as these do not report any result on egocentric datasets.
>
> **Table X:** Comparison of our approach with SOTA **Unsupervised Action Segmentation** methods. Note '-' denotes authors have not provided any data on those metrics.
> |Action Segmentation Papers| |ProceL | | | |CrossTask| |
> |:----|:---:|:---:|:---:|:-:|:---:|:---:|:---:|
> | |P|R|F1||P|R|F1|
> |JointSeqFL (2019) [a]|-|-|29.8||-|-|-|
> |Elhamifar et al. (2020) [b]|9.5|26.7|14.0||10.1|41.6|16.3|
> |Fried et al. (2020) [c]|-|-|-||-|28.8|-|
> |Shen et al. (2021) [d]|16.5|31.8|21.1||15.2|35.5|21.0|
> |Dvornik et al. (2022) [e]|-|-|-||-|-|25.3|
> |StepFormer (2023) [f]|18.3|28.1|21.9||22.1|**42**|28.3|
> |OPEL|**33.6**|**36.3**|**34.9**||**35.6**|34.8|**35.1**|
>
>
> **Temporal Alignment methods:** According to the reviewer's suggestion, we have included additional comparisons with temporal alignment techniques in **Table Y**. Specifically, we have compared our model with methods like **TCC** [g] and **LAV** [h], which incorporate temporal cycle consistency (TCC) and dynamic time warping (DTW), respectively. Other methods like **CnC** [i] uses TC3I (TCC + contrastive inverse difference moment, C-IDM) as the loss function while **GPL** [j] uses graph-based representation for temporal alignment. Our results clearly demonstrate the efficacy of our approach compared to these established methods, further validating the effectiveness of our model in maintaining temporal alignment while delivering superior PL performance. Note, in Table Y, our approach is compared with SOTA **Unsupervised Temporal alignment** methods for only egocentric (first-person) datasets, as these do not report any result on third-person datasets.
>
> **Table Y:** Comparison of our approach with existing **Unsupervised Temporal Alignment** methods.
> |Temporal Alignment Methods | CMU-MMAC| | | | MECCANO| | | |EPIC-Tent| | | |EGTEA-GAZE+| | | |PC Assembly| | | |PC Disassembly| | |
> |:----|:---:|:---:|:---:|:---:|:---:|:---:|:---:|:---:|:---:|:---:|:---:|:---:|:---:|:---:|:---:|:---:|:---:|:---:|:---:|:---:|:---:|:---:|:---:|
> | |Precision|F1|IoU| |Precision|F1|IoU| |Precision|F1|IoU| |Precision|F1|IoU| |Precision|F1|IoU| |Precision|F1|IoU|
> |TCC [g]|18.5|19.7|9.5| |15.1|17.9|8.7| |14.2|14.9|7.8| |17.5|19.7|8.8| |19.9|21.7|11.6| |22.0|23.4|12.2|
> |LAV (DTW) [h]|20.6|21.1|9.4| |14.6|17.4|7.1| |15.2|15.8|8.3| |17.4|19.1|8.0| |21.5|22.7|11.7| |26.4|26.5|12.9|
> |LAV + TCC|18.8|19.7|9.0| |13.4|15.6|7.3| |16.0|16.7|8.5| |16.4|18.6|7.5| |21.6|21.1|10.8| |21.0|24.3|12.3|
> |CnC(TC3I) [i]|21.6|22.7|11.1| |15.5|18.1|7.8| |17.1|17.2|8.3| |19.6|21.7|9.5| |25.0|25.1|12.8| |28.4|27.0|14.8|
> |GPL [j]|30.3|31.7|17.9| |18.8|20.7|10.0| |17.9|19.8|9.1| |23.8|27.1|16.0| |27.1|27.5|15.2| |28.1|26.7|15.2|
> |OPEL|**32.8**|**36.5**|**18.8**| |**28.9**|**39.2**|**20.2**| |**18.8**|**20.7**|**10.6**| |**24.3**|**29.5**|**13.2**| |**32.5**|**33.7**|**17.9**| |**29.6**|**32.2**|**16.7**|
>
> Thanks again for the suggestions to strengthen our paper, looking forward to your kind consideration.
>
> **Refs.:**
>
> [a] Elhamifar *et al.* Unsupervised Procedure Learning via Joint Dynamic Summarization, ICCV 2019.
>
> [b] Elhamifar *et al.* Self-supervised multi-task procedure learning from instructional videos, ECCV 2020.
>
> [c] Fried *et al.* Learning to segment actions from observation and narration, ACL 2020.
>
> [d] Shen *et al.* Learning to segment actions from visual and language instructions via differentiable weak sequence alignment, CVPR 2021.
>
> [e] Dvornik *et al.* Flow Graph to Video Grounding for Weakly-Supervised Multi-step Localization, ECCV 2022.
>
> [f] Dvornik *et al.* StepFormer: Self-supervised Step Discovery and Localization in Instructional Videos, CVPR 2023.
>
> [g] Dwibedi *et al.* Temporal cycle-consistency learning, CVPR 2019.
>
> [h] Haresh *et al.* Learning by aligning videos in time, CVPR 2021.
>
> [i] Bansal *et al.* My view is the best view: Procedure learning from egocentric videos. ECCV 2022.
>
> [j] Bansal *et al.* United we stand, divided we fall: Unitygraph for unsupervised procedure learning from video. WACV 2024.
>
> [k] Kumar *et al.* Unsupervised Action Segmentation by Joint Representation Learning and Online Clustering. CVPR, 2022.

---

> > ### Author Response · Authors · 2024-08-10
> > **Discussion period**
> >
> > Dear Reviewer, thanks again for your feedback, hope you consider our responses. Please let us know if you have any further queries, looking forward to the discussion.

---

### Official Review · Reviewer_Yy5N · 2024-07-12

**Soundness:** 3
**Presentation:** 2
**Contribution:** 3
**Rating:** 5
**Confidence:** 3

**Summary:**

The authors propose a novel approach for procedure learning leveraging optimal transport, OPEL. OPEL integrates optimality and temporal priors, and incorporates a novel inter-video contrastive loss. OPEL achieves significant improvements on egocentric and third-person benchmarks.

**Strengths:**

1. An interesting and relevant topic regarding the treatment of procedure learning as an optimal transport problem.
2. The authors thoroughly consider deviations in real-world sequences, including background frames, redundant frames, and non-monotonic frames.
3. OPEL achieves excellent results across several benchmarks.

**Weaknesses:**

1. The presentations of the paper needs improvement, especially in Section 3, where an excessive number of formulas obstruct readability.
2. Some figures are unclear, such as Figure 1. Are V1, V2, V3, V4 from four different videos, or do they appear to be from the same video? Additionally, what distinguishes V3 from V4, where one represents background frames and the other represents redundant frames?

**Questions:**

1. I still have some doubts regarding the difference between procedure learning and action segmentation. I believe their final output forms are same. From related work, I believe that action segmentation involves classifying each frame, while procedure learning only needs to identify key steps. Is this correct? Could you please explain in detail?
2.I also feel that the paper doesn't clearly present the problem formulation, including the input and output, which makes it somewhat confusing to understand. Does the output video contain a fixed number of K key steps? According to the paper, setting K to 7 is optimal. However, based on the dataset, the average number of steps for each task may be greater than or less than K. When it exceeds K, are some steps merged? And what happens when it's less than K? Is it reasonable to pre-define K?
3. Could you provide more results of ablation training on K for additional tasks?

**Limitations:**

Yes

---

> ### Author Rebuttal · Authors · 2024-08-02
>
> Thanks for your encouraging feedback.  Please find answers to your specific comments:
>
> **Presentation:** We agree that the equations are too dense in Section 3, and the formulations can be simplified. We will update accordingly as per the reviewer’s suggestion during revision.
>
> The V1 to V4 in Fig. 1 are from four different videos, while the frames from the same video are shown in x-axis. We apologize for the ambiguity of background frames vs redundant frames; the Fig.1 has been updated as **Fig. R1** (rebuttal pdf). Both refer to frames not directly related to any key-step (hence, do not belong to any cluster) and thus are clustered together as a background cluster (**Fig. R1**, right). So, we remove the term redundant frames for clarity.
>
> **PL vs AS:** Procedure learning (PL) and action segmentation (AS) are related but not the same. PL, when applied to a set of videos of the same task, involves 2 primary steps: (a) assigning each frame to 1 of the *k* key-steps (including background elements), (b) determining the logical sequence of the key-steps to complete the task. As illustrated in Fig. 1, PL addresses multiple videos of a given task, enabling the identification of repetitive key-steps across these videos [a, b]. In contrast, AS [c] focuses on a single video, thereby lacking the ability to discern repetitive key-steps across videos. Additionally, AS does not consider the sequence of individual events, which is often crucial for accurately identifying procedures. For instance, AS fails to capture the variations in the order of key-steps, such as those observed between V1 and V2 in Fig. 1.
>
> In our work, the **OPEL** loss is used to train a representation learning algorithm to obtain an embedding space where similar frames are located close by. Then, as described in the ‘Clustering and Key-step Ordering’ part of page 6 of the main paper, these key-steps are clustered, and their sequential order is determined. A PyTorch function (with a toy example) is provided in **codeblock R1** of the rebuttal pdf to illustrate this process of finding the sequential order (part b from the definition of PL given above) of the frame-wise key-step predictions (output from part a from the above PL definition).
>
> **Problem formulation:** Given a set of videos of the same task (e.g. making brownies), PL aims to find the constituent key-steps (e.g. break egg, mix contents, add oil, mix egg) and their sequential order (break egg --> mix egg --> add water --> add oil --> mix contents).
>
> During training: Inputs are untrimmed and unlabeled videos. We train with pairs of videos at a time (formulation detailed in Section 3 of the main paper). Output is the frames clustered into different key steps and their sequential order. For evaluation metrics, we use framewise cluster predictions, following other SOTA works [a,b,d].
>
> During inference:
> Input: a single video, $P = \left[p_1,\ p_2,\ .\ .\ .\ ,\ p_N\right]$
> Output:
> Assign each frame to a phase using trained model $f_{\theta}∶ f_{\theta}(p_i)= l_i$, where $l_i ∈ {c}^k_{c=1}$, represents the key-step corresponding to the phase of each frame $p_i$ in the video P.
>
> From all $l_i$, we determine the sequential order of the sub-tasks (eg: 0,3,2,4,1...). Note, the cluster number is arbitrary, each of them denoting a key-step. We provide a PyTorch code to illustrate this process of finding the sequential order in **codeblock R1** of the rebuttal pdf.
>
> **Choice of *k*:** In unsupervised PL, *k* is not set during training (learning of the embeddings), rather *k* is used during inference only to fix the no. of output clusters, and each frame of the video being inferred is mapped to one of those *k* clusters. Now, an output video may not contain that exact *k* number of key-steps (e.g. in making a sandwich, steps such as adding jelly, butter etc. may be present in some videos, while absent in others).
>
> We obtain best results with *k=7* (Table 7), consistent with other SOTA methods on the same datasets [a,b,d]. We agree that based on the dataset, the average number of steps for each task may be > or < *k*. If it exceeds *k*, some steps get merged, e.g., for PC Disassembly, though the ground-truth number of steps is 9, 3 are quite similar (remove hard disk, remove motherboard, remove RAM); effectively making them quite close in the feature space. As a result, choosing *k=7* results in these 3 key-steps getting merged. When the average number of key-steps becomes < *k*, each subtask gets split into multiple smaller clusters with very similar embeddings. This phenomenon is illustrated in **Fig. R2** (rebuttal pdf), where a larger k might result in several split clusters of small windows (blue and red boxes in **Fig. R2**).
>
> Note, this demarcation of subtasks (no. of clusters) varies from task to task, dataset to dataset and is subjective as some may consider semantically similar tasks (e.g. pouring oil vs water) to be same subtask, while others may consider it different. So, for best performance, *k* might be adjusted task-wise (using methods such as the elbow method, AIC criterion). However, optimizing for task-wise *k* is not the goal of this work; our contribution is on the OT-based learning and the clustering is only used as an inference post-processing step. So for fair comparison, we experiment with some reasonable values of *k* (following prior art [a,b,d]) and consistently outperform them across all *k* and all datasets with similar trends.
>
> We report **additional ablation** on varying *k* in the rebuttal pdf, **Table R1**. We find that the performance degrades as *k* is increased from $7\to10\to12\to15$, thus overall *k=7* provides the best performance. Note, our contribution is on the OT-based learning, the clustering is only used as an inference post-processing step. Overall, we achieve similar trends for varying *k* w.r.t. SOTA works [a,b,d].
>
> [a] Bansal et al. ECCV 2022
>
> [b] Bansal et al. WACV 2024
>
> [c] Kumar et al. CVPR, 2022
>
> [d] Shah et al. ICCV, 2023

---

> > ### Author Response · Authors · 2024-08-10
> > **Discussion period**
> >
> > Dear Reviewer, thanks again for your feedback, hope you consider our responses based on your comments. Please let us know if you have any further queries, looking forward to the discussion.

---

### Official Review · Reviewer_pvnX · 2024-07-12

**Soundness:** 3
**Presentation:** 2
**Contribution:** 3
**Rating:** 6
**Confidence:** 3

**Summary:**

The paper proposes an unsupervised method for procedure learning that identifies the key steps and their orders in several videos of the same task.
The paper formulates the distribution of video frames as an optimal transport (OT) problem to compute the distances between the key steps.
To handle the variations of the videos, the paper introduces a regularization with prior distribution.
The proposed method achieves SOTA on third-person and first-person video benchmarks.

**Strengths:**

- Different from prior methods having ordering constraints assumption, the paper relaxes this by formulating OT.

- The paper addresses the variation between videos and then introduces a regularization with priors to enhance the OT.

- The paper shows SOTA results in both first-person and third-person evaluations.

**Weaknesses:**

- The explanation of using priors to mitigate the variation of the videos, i.e., action speeds, non-monotonic sequences, or starting of actions, is not clear.
- Besides the ablation studies of different prior distributions, the paper should explain why Laplacian outperforms other distributions.

**Questions:**

The authors should give a clear explanation of the regularization usage as mentioned in the weakness.

**Limitations:**

The authors have mentioned the limitations.

---

> ### Author Rebuttal · Authors · 2024-08-02
>
> Thanks for your encouraging feedback.  Please find the answers to your specific comments:
>
> **Explanation of priors:** The concept of the **Optimality Prior** is crucial when dealing with video alignment, especially in challenging scenarios. When two videos are perfectly aligned (case 1 of Fig. 2B), the $\hat{T}$ becomes strictly diagonal. However, the real value of the optimality prior emerges in more complex situations, such as speed variation, non-monotonicity, etc.
>
> **Speed variation:** Consider the scenario in Fig. 2B, case 3: video 1, shown at the top, takes two frames to complete task A, while video 2, displayed on the left, completes task A in one frame. Here, both frames 1 and 2 from video 1 should be aligned with frame 1 of video 2. The optimality prior enables this alignment, as depicted by the blue curve in Fig. 2C and explained by Eqn. 2. The procedure involves finding, for each frame $i$ from video 2, the closest matching frame from video 1 based on feature similarity, even if the corresponding frame $j$ from video 1 has a different index. In Fig. 2C, for example, frame $j$ from video 1 has the highest likelihood of aligning with frame $i$ from video 2, with the assignment likelihood decaying exponentially for frames further away from $j$.
>
> The optimality prior also addresses **non-monotonic** cases (Fig. 2B, case 4). Here, though the first two frames from video 1 align with those in video 2, the third frame of video 1 reverts to an earlier task A. To correctly align this frame, it must be matched with its closest counterpart in video 2, which is frame 1 in this case. The optimality prior facilitates this by aligning non-monotonic frames that share a higher feature-wise match (similar to Fig. 2C).
>
> However, relying solely on the optimality prior is insufficient because it overlooks the temporal ordering inherent in videos, which is crucial for maintaining temporal coherence. As discussed in the regularization section on page 4 of our paper and illustrated in **Fig. R4** (rebuttal pdf), optimizing only for optimality can result in temporal incoherence. Proper alignment should account for the temporal relationship between frames, ensuring that corresponding frames in one sequence align closely with adjacent frames in the other.
>
> This necessity leads to a second critical factor: the **Temporal Prior**. Unlike the optimality prior, which seeks the best feature match regardless of temporal distance, the temporal prior promotes alignment between frames that are temporally adjacent, thereby preserving the overall temporal coherence. Similar coherence based concepts have been utilized in other temporal alignment works, such as TCC [a], and contrastive regularization [b]. This temporal prior encourages the alignment matrix to exhibit peak values along the diagonal, with diminishing values away from the diagonal; we model this with a Laplace distribution (Eqn. 3 and the red curve in Fig. 2C).
>
> Essentially, there are 2 factors in play- (i) *optimality* which tries to find the best match between frames irrespective of their temporal distance (which may result in temporally incoherent alignment), (ii) the *temporal* factor which promotes transport between nearby frames without considering their feature matching. We hypothesize (and later validate with results) that the optimal solution requires a balance between both, and therefore propose to merge these two priors addressing the above factors, as expressed in Eqn. 4. **This combined prior** ensures accurate alignment between videos considering all the factors, as further illustrated in Fig. 2D and supported by the results in Table 5 of the main paper.
>
> When the **starting of actions** is different (case 2, Fig. 2B), video 2 has a background frame to start when video 1 starts doing task A. As mentioned in page 5 of our paper, we introduce a \'virtual frame\' to handle such cases. Even in this case, the combined prior comes into play as we assign any frame to the virtual frame if the likelihood of that frame aligning with any other task-related frames from the other video falls below a predefined threshold. This phenomenon is also illustrated in Fig. 4(B) of the main paper.
>
> **Laplace distribution:** As shown in appendix Table A6 (main paper), Laplace as a prior outperforms other distributions. To analyze this, we plot the distributions in **Fig. R3** (rebuttal pdf). Note, we use the same distribution for optimality as well as temporal priors.
>
> For the *optimality prior*, the x-axis is the difference of frames in the feature space (1-d representation for illustration purposes), and the y-axis denotes the corresponding probability of alignment. We want the point representing the most likely alignment (as per $\hat{T}$) to have the highest likelihood, while the assignment probability should exponentially decay further away. The graph clearly shows that the Laplace distribution captures this behavior suitably compared to Uniform and Gaussian. Similarly, for the *temporal prior*, the x-axis denotes the temporal distance between the frames, and the y-axis denotes the corresponding probability of alignment. The graph shows that Laplace distribution facilitates alignment of the frames when they are temporally aligned (close to center), and its **long tail distribution** enables better correlation of non-monotonic frames compared to Gaussian or Uniform. As a result, as shown in **Fig. R3**, even at far away locations from the center (temporally distant frames), alignment is possible if indeed these frames have a high match feature-wise. In this case, the Laplace temporal prior provides non-zero probability to that far away frame (due to its long tail) unlike other distributions and the optimality prior gives a large score (due to feature match), resulting in improved handling of non-monotonicity.
>
> **Refs:**
>
> [a] Dwibedi *et al.* Temporal cycle-consistency learning. CVPR 2019.
>
> [b] Haresh *et al.* Learning by aligning videos in time. CVPR 2021.

---

> > ### Author Response · Authors · 2024-08-10
> > **Discussion period**
> >
> > Dear Reviewer, thanks again for your feedback. We hope you consider our responses based on your comments, looking forward to the discussion.

---

> > > ### Comment · Reviewer_pvnX · 2024-08-12
> > >
> > > Thank the authors for their responses. I am satisfied with the answers. After reading other reviews, I will raise my score to 6.

---

> > > > ### Author Response · Authors · 2024-08-14
> > > > **Thank you**
> > > >
> > > > We thank the reviewer for the consideration and are glad that the reviewer liked our responses.

---

### Official Review · Reviewer_XDhK · 2024-07-12

**Soundness:** 3
**Presentation:** 3
**Contribution:** 3
**Rating:** 7
**Confidence:** 3

**Summary:**

OPEL is a novel technique for Procedure Learning. Procedure learning is the task of finding key steps in an action (such as cooking brownie) and aligning the videos based on these key steps.
OPEL proposes to use the optimal transport distance between the two videos as the similarity metric rather than direct frame by frame mapping or assuming a strict monotone mapping.
This work also proposes two regularizers to incorporate priors about monotone mapping and about increasing the correspondence between nearby frames.
The results on both egocentric and 3rd person videos show significant improvement upon the previous SOTA tasks.
The ablation studies analyze the effect of each of the losses and regularizers.
It seems having the temporal prior and the inter-intra cross entropy are the most important factors in terms of f1 accuracy.

**Strengths:**

This is a sound and novel technique for procedure learning. Using optimal transport between frames significantly improves the results and add leniency when the frames are not exact matches temporally. This is a common happenstance in real world scenarios since not all the steps of a procedure need to happen in chronological order. Two sub procedures can be interchangeable and using optimal transport with only a regularizer on temporal monotony is can account for this.

The experiments are sufficiently evaluated on several datasets. Over all the datasets this method out performs the prior work.

The manuscript is written in a detailed fashion and the full information is provided for sake of replicability. Also the ablation study provided improves the clarity of what is important in this method.

**Weaknesses:**

There are many factors added together to make this method work. Although many of them are common in other methods as well, but the method and also the ablation seem to suggest that some of the losses may be redundant. It is not clear if the difference between some of the lines in table. 5 is statistically significant.

Page 6. is written in such a compact way that it's not easy to read and follow. The indices such as i,j can be probably dropped. Words such as temperature in the formula can be probably replaced with tau.

**Questions:**

Why do you think the number of steps makes such a large difference in table 7? going from 7 to 10 drops the performance significantly. another anomaly is 12 being worse than both 10 and 15.

**Limitations:**

not explicitly.

---

> ### Author Rebuttal · Authors · 2024-08-02
>
> Thanks for your encouraging and insightful feedback. Please find the answers to your specific comments. Also please refer to our overall rebuttal response to all reviewers and the corresponding pdf containing additional figures and tables to support our claims.
>
> **Method factors:** The reviewer correctly points out that all the components are not equally critical for OPEL’s performance. We show the contributions of each factor individually to analyze their effect on the overall result in Table 5. Comparing row 3 with row 9, we observe, the priors jointly play a critical role; as without them (row 3), the F1 and IoU scores drop by *\~5* points. Specifically, the optimality prior has a significant impact (*\~2* point), while the temporal prior affects the score by *\~1* point. Similar to the combined priors, the intra and inter-video contrastive losses together (see row 6 vs row 9) have a significant effect (*\~3.5* points) on the overall performance. The individual effect of virtual frame is negligible as it only plays a role in case of excessive background frames - a scenario that is not prevalent in most datasets. Furthermore, due to the IDM structure of $M\left(\hat{T}\right)$, the $\hat{T}$ and $\hat{Q}$ are similar by formulation. This results in $KL\left(\hat{T}\ \parallel\hat{Q}\right)$ to be already small. As a consequence, adding the KL divergence as a standalone loss component in the proposed pipeline has a minimal impact. Overall, while some loss components may have a smaller individual impact, they do contribute to performance improvements, even if incrementally. Therefore, our proposed approach incorporates all of them to achieve the best possible results. We are encouraged with the reviewer's note - the provided ablation study improves the clarity of what is important in this method.
>
> **Writing:** The equations are an integral part in explaining the proposed approach and the different loss terms. But we agree that the page 6 is too dense, and the formulations can be simplified to improve readability. We will update the notations and improve the overall writing as per the reviewer’s suggestion during revision.
>
> **Effect of k:** Note, we obtain best results with *k*=7 (Table 7), and performance drops sharply as *k* goes from 7 to 10. This observation is consistent with all the other SOTA methods on the same datasets [a, b, c]. We hypothesize that *k*=7 works best as it is the optimal number of clusters considering the average number of distinct key-steps (subtasks) of the datasets. For example, for PC Disassembly, although the ground-truth (GT) number of steps is 9, 3 steps are quite similar (remove hard disk, remove motherboard, remove RAM), effectively making them quite close in the feature space. This results in *k*=7 being a better estimation of the cluster number with distinct steps. Note, this demarcation of subtasks (hence, number of clusters) is subjective and varies from dataset to dataset as well as from task to task;  as some may consider semantically similar tasks (e.g. pouring oil vs water) to be one subtask, while others may consider it different. As *k* becomes larger than the actual distinctive number of clusters, each subtask gets split into multiple clusters with very similar embeddings, which upon comparison with GT leads to inferior results. This phenomenon is illustrated in **Fig. R2** (see rebuttal pdf), where a larger *k* might result in several erroneous clusters with very small windows (blue and red boxes in **Fig. R2**). This leads to large fluctuations (jittery predictions) within a single GT phase, thus deteriorating the overall performance.
>
> Secondly, the anomalous trend (*k* = 12 being slightly worse than 10 and 15) might be a dataset (PC Assembly) specific issue and not unique to our approach, as a similar trend for this dataset has been reported in [a, b]. However, for PC Disassembly and other datasets (reported as additional results in the rebuttal pdf, **Table R1**), we consistently find that the performance degrades as *k* is increased from 10 $\to$ 12 $\to$ 15. Note, that our contribution is on the optimal transport (OT)-based representation learning, the clustering is only used as an inference post-processing step. Overall, we achieve similar trends with respect to the SOTA works but consistently outperform them across all *k* and datasets.
>
> **Refs.:**
>
> [a] Bansal *et al.* My view is the best view: Procedure learning from egocentric videos. ECCV 2022.
>
> [b] Bansal *et al.* United we stand, divided we fall: Unitygraph for unsupervised procedure learning from video. WACV 2024.
>
> [c] Shah *et al.* Steps: Self-supervised key step extraction and localization from unlabeled procedural videos. ICCV, 2023.

---

> > ### Comment · Reviewer_XDhK · 2024-08-08
> > **Acknowledged**
> >
> > Thank you for your response. In terms of statistical significance, what is the standard deviation for a set of 5 runs for example?

---

> > > ### Author Response · Authors · 2024-08-09
> > > **Consistent results across multiple runs**
> > >
> > > Thanks for your response. In our paper, we reported just the **mean values** obtained over multiple runs and not the standard deviations as the results did not vary significantly. Also, previous SOTA works [a, b, c] do not report the standard deviation across runs either. However, as per the reviewer's suggestion, we report the results of the mean ± standard deviation (SD) over 5 separate runs in **Table A** below. Note, in all cases, we find that the SD is quite low, and we get consistent results over the multiple runs, further demonstrating the statistical significance of the results.
> > >
> > >
> > > **Table A:** Results showing mean ± SD over 5 runs for all the datasets
> > >
> > > | Dataset         | F1 (mean ± SD)    | IoU (mean ± SD)    |
> > > |:----------------|:-----------------:|:------------------:|
> > > | CMU-MMAC        | 36.5 ± 0.138      | 18.8 ± 0.106       |
> > > | EGTEA-GAZE+     | 29.5 ± 0.147      | 13.2 ± 0.145       |
> > > | MECCANO         | 39.2 ± 0.319      | 20.2 ± 0.258       |
> > > | EPIC-Tents      | 20.7 ± 0.165      | 10.6 ± 0.101       |
> > > | PC Assembly     | 33.7 ± 0.311      | 17.9 ± 0.184       |
> > > | PC Disassembly  | 32.2 ± 0.317      | 16.9 ± 0.203       |
> > > | ProceL          | 34.9 ± 0.095      | 21.3 ± 0.037       |
> > > | CrossTask       | 35.1 ± 0.142      | 21.5 ± 0.111       |
> > >
> > > Please let us know if you have any further queries. Thanks again and hope you reconsider your score based on our responses.
> > >
> > > **Refs.**:
> > >
> > > [a] Bansal et al. My view is the best view: Procedure learning from egocentric videos. ECCV 2022.
> > >
> > > [b] Bansal et al. United we stand, divided we fall: Unitygraph for unsupervised procedure learning from video. WACV 2024.
> > >
> > > [c] Shah et al. Steps: Self-supervised key step extraction and localization from unlabeled procedural videos. ICCV, 2023.

---

### Author Rebuttal · Authors · 2024-08-02

We thank all the reviewers for their insightful comments and feedback. We are encouraged that the reviewers like the soundness and novelty of our approach for procedure learning along with comprehensive evaluation (Reviewer XDhK), and enhancement of results over current state-of-the-art (SOTA) works (Reviewer pvnX). Additionally, Reviewer Yy5N finds our work interesting and relevant as we thoroughly consider deviations in real-world sequences, including background and non-monotonic frames. Lastly, Reviewer 3Xtn appreciates the novelty of our work and its SOTA performance.

The reviewers have also raised some concerns that we have addressed in their individual rebuttal responses. Furthermore, as per the reviewers' suggestions, we have added additional relevant figures and results to support our claims in the attached 1 page pdf and referred to them in our responses. Specifically,

1. Figure R1 is a modified version of Figure 1 of the main paper.
2. Figure R2 illustrates the qualitative effect of varying *k* for clustering the key-steps.
3. Figure R3 shows the reasoning for the choice of Laplace distribution as a prior over other distributions.
4. Figure R4 emphasizes of the importance of both the optimality and temporal priors to combat real-world non-idealities like non-monotonic frames, speed variation, etc., while maintaining temporal coherence.
5. Table R1 shows additional ablation study on the effect of *k* on MECCANO and EPIC-Tents datasets.
6. Codeblock R1 depicts a Pytorch function to determine the sequential ordering of tasks from frame-wise key-step predictions.

In general, we agree that the formulations in Section 3 can be simplified to improve readability. We will update accordingly as per the reviewers' suggestions during the camera-ready version, if accepted. Please note, in our rebuttal responses, Fig./Table X (e.g. Fig. 2/Table 5) refers to the main submitted manuscript, while **Fig./Table RX** (e.g. Fig. R2/Table R1) refers to the **rebuttal pdf** attached herewith.

Once again, we sincerely appreciate your time and consideration. Please let us know if you have any further queries. We look forward to your responses.

---

### Public Comment · ~Quoc-Huy_Tran2 · 2025-12-01
**This paper shouldn't have been accepted. The authors are doing the same "trick" again with ICLR 2026.**

This paper has heavily copied existing works and has very limited novelty. It is merely a copy of previous works in the same self-supervised video alignment and procedure learning topics, i.e., VAVA [1] (for self-supervised video alignment) and CnC [2] (for clustering and key-step ordering). In Sec. 3:

* **"Optimal Transport Formulation"**, **"Background and Redundant Frames"**, **"Training Methodology"**, **"Contrastive Regularization"** are exactly copied from VAVA [1].

* **"Regularization with Priors"** is copied from VAVA [1]. Temporal priors and Optimality priors were proposed in VAVA [1]. The only difference here is the use of Laplace function in Eq 2 and 3 in OPEL vs. Gaussian function in Eq 4 and 6 in VAVA [1], but it is a minor novelty.

* **"Clustering and Key-step Ordering"** is exactly copied from CnC [2].

The authors make it sound like the above modules are proposed by them by 1/ using new names for existing modules, and 2/ missing citations of the original works and missing discussions on how their modules are different from those in the original works in Sec. 3.

The authors submit this paper to a Machine Learning conference, so it has less chances to be assigned to reviewers who work on video alignment/action segmentation/procedure learning. All of the competing methods in Tabs. 1, 2, 3 are published in Computer Vision conferences like ICCV/CVPR/ECCV/WACV.

Since the authors have been "succeeded" with this NeurIPS 2024 paper, they are now applying the same "trick" again with ICLR 2026. Their new submission (you can find by searching for the second author on Google Scholar) copies VAVA [1]/OPEL [2], and my VAOT [3]/RGWOT [4] works, with very limited and questionable novelty. I have added my detailed comments on their OpenReview submission.

[1] Liu et al. Learning to align sequential actions in the wild. CVPR 2022. (VAVA)

[2] Bansal et al. My view is the best view: Procedure learning from egocentric videos. ECCV 2022. (CnC)

[3] Ali et al. Joint self-supervised video alignment and action segmentation. ICCV 2025. (on ArXiv since March 2025) (VAOT, VASOT)

[4] Mahmood et al. Procedure learning via regularized gromov-wasserstein optimal transport. WACV 2026. (on ArXiv since July 2025) (RGWOT)

P/S: I am Quoc-Huy Tran, who is an expert in self-supervised video alignment (e.g., LAV at CVPR 2021, LA2DS at ECCV 2024, VAOT at ICCV 2025), self-supervised action segmentation (e.g., TOT at CVPR 2022, UFSA at WACV 2024, VASOT at ICCV 2025), and self-supervised procedure learning (e.g., RGWOT at WACV 2026). Two of my works are cited in this paper. I have been aware of this work since March 2025 when I started working on procedure learning. I didnt know I could add public comments until just now.

---

### Public Comment · ~Sayeed_Shafayet_Chowdhury2 · 2025-12-03
**[1/2] Rebuttal to Unfounded Claims on Attribution and Clarifying Novelty in OPEL**

We want to state unambiguously that we **categorically reject** the allegation that OPEL merely copies VAVA and CnC without citation or by renaming modules, thereby misrepresenting our own contributions. These accusations are **not supported by the text of the paper** and repeatedly misstate what we claim, cite, and actually contribute, often ignoring explicit attributions and methodological differences that are already spelled out in the Introduction, Related Work, and Section&nbsp;3. We detail the relevant points below.

## 1. Problem setting: unsupervised procedure learning vs. self-supervised video alignment

OPEL is a method for unsupervised procedure learning (PL): given multiple videos of a task, we aim to (i) localize key-steps and (ii) recover their logical ordering without any frame-level labels. This PL setting is explicitly defined in the Introduction and Related Work and contrasted with action segmentation and with generic video alignment / representation learning.

In the *Video Alignment* paragraph of Related Work (page&nbsp;3), we explicitly discuss VAVA&nbsp;[1] as the closest OT-based video alignment method and state that:

- VAVA uses OT for self-supervised video alignment,
- its evaluation setup is supervised fine-tuning for action segmentation, and
- as a result, it does **not** address temporal localization of key-steps or their ordering, unlike our PL setting.

We also state there that our **modeling of priors using Laplace distributions** and our **inter-video contrastive loss formulation** are different from [1] (VAVA).

Thus, we explicitly position VAVA as prior work on OT-based video alignment for representation learning, and OPEL as an OT-guided framework for unsupervised procedure learning, which must both align and segment key-steps and recover their ordering across videos.

## 2. Attribution to VAVA in Section 3 (priors, background frames, training, contrastive terms)

In Section&nbsp;3 (“OPEL Framework”), the components that have been alleged to be “copied” are in fact explicitly attributed to VAVA where appropriate:

### Regularization with priors

In the “Regularization with Priors” subsection (page&nbsp;4), we introduce the optimality and temporal priors and write that we

> “… enhance the OT formulation by incorporating two specific priors addressing the above factors [46] (VAVA)”,

i.e., we explicitly motivate this part by VAVA. We then **change the modeling**: we model the priors as **Laplace distributions** (Equations&nbsp;2–3), which leads to a different analytical form and behavior than the **Gaussian-mixture prior** used in VAVA. In particular, the Laplace distribution facilitates alignment of frames when they are temporally aligned (near the center), and its long-tailed behavior enables better correlation of non-monotonic frames compared to Gaussian or uniform priors; further details were provided during our submission.

### Background and redundant frames

In “Background and Redundant Frames” (page&nbsp;5), we state that

> “integrate an additional ‘virtual frame’ within the transport matrix, following [46] (VAVA)”

to handle background and redundant frames. Thus, we already clearly credit VAVA for the virtual-frame idea.

### Training methodology

In “Training Methodology” (page&nbsp;5), we explicitly mention that we are

> “Following [46] (VAVA)…”

when defining the IDM-style regularizers. At the same time, Equations&nbsp;7–9 and the derivation of the optimal transport matrix $\hat T$ in OPEL (page&nbsp;6) are **developed independently beyond VAVA’s formulation**, as VAVA does not discuss this. The detailed derivation of this optimal $\hat T_{\lambda_1,\lambda_2}$ is provided in Appendix&nbsp;A.1 of OPEL (developed independently beyond VAVA), and our provided code is based on that. The VAVA submission does not provide any associated implementation, to the best of our knowledge.

### Contrastive regularization

For the **intra-video** part, we follow prior work on C-IDM and explicitly cite [41, 1] (page&nbsp;6). These are **not** contributions from VAVA; VAVA itself cites these prior works (see page&nbsp;6, above Eq.&nbsp;14 of VAVA&nbsp;[1]). For the **inter-video** part, we introduce a OT-guided contrastive term that uses the best and worst matches according to $\hat T_{\lambda_1,\lambda_2}$ to define “positive” and “negative” distances and then applies a cross-entropy loss. This differs from VAVA’s contrastive formulation and is tailored to our PL objective.

Therefore, to say that OPEL “misses citations” of prior work in Sec.&nbsp;3 or that the modules are simply renamed from VAVA is **completely unfounded, unjustified, and misleading**. We use the same terminology (don't rename), and we **explicitly cite prior art** wherever we adopt or adapt existing ideas. Our contribution is to **adapt and extend** them into a new regularized OT objective ($L_{OPEL}$) with distinct derivations (Equations&nbsp;7–9 and Appendix&nbsp;A.1) suitable for unsupervised PL.

---

> ### Public Comment · ~Sayeed_Shafayet_Chowdhury2 · 2025-12-03
> **[2/2] Rebuttal to Unfounded Claims on Attribution and Clarifying Novelty in OPEL**
>
> ## 3. Attribution to CnC for clustering and key-step ordering
>
> CnC&nbsp;[2] is a central prior work and baseline in our paper:
>
> - In our “Clustering and Key-step Ordering” subsection (page&nbsp;6), we state that, for ordering, we compute normalized time indices for frames
>   > “following [1],”
>   where [1] in that section is CnC, and then sort clusters by the mean normalized time to obtain the key-step sequence.
>
> We therefore do **not** claim graph-cut clustering or time-based ordering as our contributions; these are standard PL tools from CnC and related work, with explicit citations in the method section. In fact, the discussion of clustering and key-step ordering in RGWOT&nbsp;[4] (see Section 3.2, page 5 of [4]) is very similar to ours in OPEL (note that OPEL precedes RGWOT). It is **inconsistent and unfair** to use a similar level of attribution in one’s own paper and then characterize the same practice as “copying” when it comes to other people’s work. This sort of attitude shifts the tone from any genuine attempt at constructive feedback or legitimate scientific critique to a needlessly hostile, disparaging, and intimidating commentary.
>
> Furthermore, we directly compare our results with CnC and show that the OPEL-based objective combined with graph-cut clustering performs **significantly better** than what CnC&nbsp;[2] proposed.
>
> ## 4. Contributions of OPEL
>
> We view OPEL’s main contributions as:
>
> 1. **Formulating unsupervised PL as an OT problem** over frame embeddings across videos, relaxing strict monotonicity assumptions while still aligning semantically similar key-steps.
> 2. **Integrating Laplace-modeled temporal and optimality priors** into a constrained OT objective ($L_{OPEL}$) with IDM-style and KL regularization, tailored to PL, together with a virtual frame to handle background/redundant frames.
> 3. **Introducing an OT-guided inter-video contrastive loss** to prevent trivial temporal solutions and encourage consistent alignment of key-step embeddings across videos.
> 4. **Demonstrating that this OT-guided PL framework yields substantial improvements** in IoU and F1 over existing PL methods, including CnC, on EgoProceL, ProCeL, and CrossTask.
>
> None of these claims assert that we invented OT, temporal priors, virtual frames, or graph cuts. Rather, the contribution lies in **how these components are adapted and combined** for unsupervised procedure learning and in the **empirical gains** this provides.
>
> ## 5. On venue choice
>
> Regarding the remark that we submitted to NeurIPS so that the work would have “less chances to be assigned” to video alignment / procedure learning reviewers: we respectfully disagree. NeurIPS hosts an active vision and sequence-learning community, and reviewer assignment is handled by the program committee; authors do not control which specific reviewers see their papers.
>
> Sincerely,
>
> The Authors
>
> ## References
>
> [1] Liu et al., *Learning to align sequential actions in the wild.* CVPR 2022. (VAVA)
>
> [2] Bansal et al., *My view is the best view: Procedure learning from egocentric videos.* ECCV 2022. (CnC)
>
> [3] Ali et al., *Joint self-supervised video alignment and action segmentation.* ICCV 2025 (on arXiv since March 2025). (VAOT, VASOT)
>
> [4] Mahmood et al., *Procedure learning via regularized Gromov–Wasserstein optimal transport.* WACV 2026 (on arXiv since July 2025). (RGWOT)

---

### Public Comment · ~Quoc-Huy_Tran2 · 2025-12-04
**[1/2] Minor novelty/contribution and Incorrect claim of inter-video contrastive loss**

Thank you for the responses. Let me address them below.

1. Problem setting.

**Procedure learning = video alignment + clustering and key-step ordering**. However, clustering (via graph cut) and key-step ordering (simply sorting by the average times) are copied from CnC and that has become a standard for all procedure learning methods. Thus, applying CnC for clustering and key-step ordering has no novelty/contribution here. Also, you are applying VAVA for video alignment, and VAVA is proposed for video alignment too. Thus, I also find it hard to see any novelty/contribution in applying VAVA for video alignment here. As I mentioned before, I do see the **minor novelty/contribution** in using Laplace function, instead of Gaussian function as in VAVA.

You are still claiming that **the inter-video contrastive loss is proposed by you, which is incorrect**!!!   Please have a look at the VAVA paper on Open Access, see Eq 15 and the paragraph containing it, “In our approach, in addition to using an intra-video contrastive loss term, we introduce an **optimal transport guided inter-video contrastive loss** to regularize the training process. In particular, we propose to contrast video pairs based on their similarity given by optimal transport…”.

In summary, the only minor novelty/contribution that I can see here is using the Laplace function and doing evaluations on procedure learning datasets. They **may be enough for a second-tier Computer Vision conference** such as WACV. But they **are not enough for a top-tier Machine Learning conference** like NeurIPS.

I have had submissions rejected and accepted by NeurIPS and several other top-tier conferences such as ICCV/CVPR/ECCV, so I have some sense of the **"bar" for top-tier conferences**. I have never had issues with others' works. In fact, I **welcome new works** that improve or build upon my works, since that increases the impacts of my works (there are **230+ citations** of my video alignment/action segmentation/procedure learning works). This is my **first time** adding public comments. I am hoping for a fair research environment, without **abuses/shortcuts**.

2. Attribution to VAVA in Section 3.

What I mean is lacking citations/differences with VAVA and CnC in Sec 3. However, most importantly, citing or not citing **does not change the fact** that you are using VAVA (with the minor modification of Laplace) for video alignment here, and VAVA is proposed for video alignment. Thus, OPEL is merely a combination of VAVA (with the minor modification of Laplace) and CnC for the **same purposes** (video alignment + clustering and key-step ordering) that they are proposed for. Also, I believe reimplementing an existing work or providing a detailed derivation of an existing work **is not considered as a novelty/contribution**.

C-IDM is proposed by my LAV work (CVPR 2021). Thus, VAVA does not claim it is their method.

---

> ### Public Comment · ~Quoc-Huy_Tran2 · 2025-12-04
> **[2/2] Minor novelty/contribution and Incorrect claim of inter-video contrastive loss**
>
> 3. Attribution to CnC for clustering and key-step ordering.
>
> CnC has become the standard for clustering and key-step ordering and is used in several procedure learning works, including OPEL and my RGWOT work. However, the **main/huge difference** here is that my RGWOT work (built on my VAOT work) is published in the **second-tier Computer Vision conference (WACV)**, whereas OPEL (built on VAVA) is published at the **top-tier Machine Learning conference (NeurIPS)**.
>
> Frankly speaking, I would have **let all (this and that) go** if they are published/submitted to **second-tier venues**, since I strongly believe their novelty/contribution is **not sufficient for top-tier venues**. That is also why I submitted my RGWOT work to WACV (second-tier venue), due to its minor novelty/contribution. RGWOT is built on my own VAOT work (I could have submitted RGWOT to the top-tier TPAMI journal instead), while OPEL is built on VAVA (not your work).
>
> 4. Contributions of OPEL.
>
> **Formulating unsupervised PL as an OT problem.** No, OT is only used for video alignment (VA) in OPEL, so this should be "Formulating unsupervised VA as an OT problem", which is already VAVA's contribution (not yours).
>
> **Integrating Laplace-modeled temporal and optimality priors.** Yes, replacing Gaussian with Laplace is a contribution, but not a significant one.
>
> **Introducing an OT-guided inter-video contrastive loss.** No, you are again claiming VAVA's contribution as yours. Please see my comment at the top.
>
> **Demonstrating that this OT-guided PL framework yields substantial improvements.** Yes, evaluating on procedure learning datasets is a contribution, but not a major one.
>
> 5. On venue choice.
>
> I respectfully disagree. It is **well known** in the community that both venue choice and primary area **matter** a lot, since reviewers and ACs are selected based on them. For example, I have been a reviewer for CVPR/ICCV/ECCV/WACV but have never been a reviewer for NeurIPS/ICLR/ICML (I was invited a few times but I refused since my expertise matches CV more than ML). All the competing methods in Tabs. 1, 2, 3 are published in CVPR/ICCV/ECCV/WACV, so I believe their authors are mostly reviewers/ACs for CVPR/ICCV/ECCV/WACV too. Your works will also have **better exposure/visibility** at Computer Vision conferences since procedure learning is essentially a Computer Vision topic and its targeted audience is Computer Vision researchers/engineers. The OT in OPEL (and VAVA) with all the priors and regularizations is **designed specifically** for video alignment, so it is not convenient to apply it in other domains.
>
> In addition, procedure learning is a **very weird/strange topic**. I only knew about it in March 2025, even though I have been working on video alignment for years. I was disappointed with how much overlap procedure learning has with video alignment. One can just take any video alignment approach (maybe with some small changes) and then add clustering and key-step clustering from CnC, but to submit that to a top-tier conference doesn't make sense due to its limited novelty/contribution.
>
> My comments have nothing to do with my company or my papers. **Several new** video alignment/action segmentation/procedure learning papers are published **every year**, e.g., five action segmentation methods are presented at ICCV 2025 alone. This is my **first time** adding public comments. I am hoping for a fair research environment, **preventing** abuses/shortcuts from **happening (again)**.
>
> Sincerely,
>
> Quoc-Huy Tran
>
> P/S: I read this tweet (https://x.com/diyerxx/status/1994042370376032701?s=20) a few days ago, and feel sympathized with it, especially the last paragraph, i.e., "This whole experience drained a lot of my time, energy, and emotion — especially because accusing others of bad data requires extra caution. I’m sharing this in hopes that the ML community remains vigilant and pushes back against this kind of **sloppy, low-quality, and irresponsible** behavior before it misleads people and wastes collective effort."
>
> I am attending NeurIPS 2025 which has 5,300 papers (vs 2,700 at ICCV) but there is **0** video alignment/action segmentation/procedure learning papers at NeurIPS 2025 (vs **5** at ICCV)!!!

---

### Public Comment · ~Quoc-Huy_Tran2 · 2025-12-23
**A PhD student at a well-known action segmentation lab shared the same opinion regarding the ICLR submission**

A PhD student at a well-known action segmentation lab, who I don't know in person, reached out to me via LinkedIn yesterday and shared the same opinion regarding the ICLR submission. Please see the screenshot here https://drive.google.com/file/d/1D-Eb44lEEE6r0KcylGPo9v3kmRo3Tlxt/view?usp=sharing. I have shared the unanonymized screenshot with the PCs since the public comment period for the ICLR submission has been passed. I believe once video alignment/action segmentation/procedure learning researchers become aware of this NeurIPS work and that ICLR submission, they will raise their voice.

Also, the authors of the ICLR submission were silent for the entire 5 days (from Nov 27 to Dec 2) since my last comment on the ICLR submission and then posted their last comment just a few hours before the public comment period of the ICLR submission ended, while I was attending NeurIPS 2025 (from Dec 2 to Dec 8) so it is almost impossible for me to reply on time. I am also not among the "official" reviewers of the ICLR submission so I didn't receive a notification when the authors posted their last comment. Thats why I addressed some of their last comment below as well.

---

### Public Comment · ~Quoc-Huy_Tran2 · 2026-01-26
**Thank you very much ACs for rejecting the ICLR submission**

Dear ACs,

I greatly appreciate your efforts for **rejecting the ICLR 2026 submission** (**REALIGN: Regularized Procedure Alignment with Matching Video Embeddings via Partial Gromov-Wasserstein Optimal Transport**) and maintaining a fair research environment **without abuses/shortcuts/plagiarisms** (claiming existing works as contributions --- see comments below and in the ICLR submission).

I strongly urge The Authors **not to merely re-title and re-submit** that work to any other top-tier machine learning conferences such as ICML. Instead, they should **improve its novelty/contribution** and re-submit it to second-tier Computer Vision conferences such as WACV, where it will be reviewed properly and have better exposure/visibility.

Best regards,

The ICLR 2026 submission: https://openreview.net/forum?id=kop52LaSAB

The ArXiv version: https://arxiv.org/abs/2509.24382

---

### Decision · Program_Chairs · 2024-09-25

**Decision:**

Accept (poster)

**Comment:**

This paper received active discussions between authors and reviewers, with the outcome that the reviewer reached an accept consensus (5, 5, 6, 7). The reviewers appreciate the novelty of the method, the new insights that come with bringing optimal transport in the video recognition domain, and the strong empirical performance. The reviewers have pointed out limitations when it comes to writing/explanations as well as the overlap with action segmentation. In the rebuttals, the authors have performed additional experiments with comparisons on action segmentation, which further strengthens the paper. The AC finds that the strengths of the paper outweigh the limitations given by the reviewers, especially after the rebuttal stage. Hence the recommendation to accept.